# Loss of MGA repression mediated by an atypical polycomb complex promotes tumor progression and invasiveness

Haritha Mathsyaraja[1], Jonathen Catchpole[1], Brian Freie[1], Emily Eastwood[2], Ekaterina Babaeva[1], Michael Geuenich[1], Pei Feng Cheng[1], Jessica Ayers[3], Ming Yu[3], Nan Wu[2], Sitapriya Moorthi[2], Kumud R Poudel[1], Amanda Koehne[4], William Grady[3,5], A McGarry Houghton[2,3], Alice H Berger[2], Yuzuru Shiio[6], David MacPherson[2]*, Robert N Eisenman[1]*

[1]Basic Sciences Division, Fred Hutchinson Cancer Research Center, Seattle, United States; [2]Human Biology and Public Health Sciences Divisions, Fred Hutchinson Cancer Research Center, Seattle, United States; [3]Clinical Research Division, Fred Hutchinson Cancer Research Center, Seattle, United States; [4]Comparative Pathology, Fred Hutchinson Cancer Research Center, Seattle, United States; [5]Department of Medicine, University of Washington School of Medicine, Seattle, United States; [6]Greehey Children's Cancer Research Institute, The University of Texas Health Science Center, San Antonio, United States

*For correspondence:
dmacpher@fredhutch.org (DMP);
eisenman@fhcrc.org (RNE)

**Abstract** MGA, a transcription factor and member of the MYC network, is mutated or deleted in a broad spectrum of malignancies. As a critical test of a tumor suppressive role, we inactivated *Mga* in two mouse models of non-small cell lung cancer using a CRISPR-based approach. MGA loss significantly accelerated tumor growth in both models and led to de-repression of non-canonical Polycomb ncPRC1.6 targets, including genes involved in metastasis and meiosis. Moreover, MGA deletion in human lung adenocarcinoma lines augmented invasive capabilities. We further show that MGA-MAX, E2F6, and L3MBTL2 co-occupy thousands of promoters and that MGA stabilizes these ncPRC1.6 subunits. Lastly, we report that MGA loss also induces a pro-growth effect in human colon organoids. Our studies establish MGA as a bona fide tumor suppressor in vivo and suggest a tumor suppressive mechanism in adenocarcinomas resulting from widespread transcriptional attenuation of MYC and E2F target genes mediated by MGA-MAX associated with a non-canonical Polycomb complex.

## Introduction

Malignant progression often results from the de-regulation of factors that drive normal developmental programs. Given the critical role of transcription factors in the control of development and in driving cellular functions, it is not surprising that transcriptional regulators can function as potent tumor suppressors or oncogenes. Arguably, one of the most well-characterized oncogenic transcription factors is MYC, which is also essential for normal development (for reviews, see *Dang and Eisenman, 2014*). Frequently altered across a broad spectrum of malignancies, MYC orchestrates transcriptional programs promoting metabolic re-wiring and ribosome biogenesis, as well as target gene independent transcriptional processes, ultimately facilitating rampant proliferation and tumor progression (reviewed in *Baluapuri et al., 2020*; *Stine et al., 2015*). A large body of evidence suggests that MYC does not function in isolation: it is part of a larger network of transcription factors that cooperate with or antagonize MYC activity (reviewed in *Carroll et al., 2018*). All members of this extended

MYC network are basic-helix-loop-helix-leucine zipper (bHLHZ) transcription factors that recognize E-box motifs and distinct aspects of chromatin structure in thousands of genes. Within the network MLX and MondoA can cooperate with MYC to promote tumorigenesis in MYC amplified cancers (*Carroll et al., 2015*). On the other hand, the MXD 1–4, MNT, and MGA proteins oppose MYC transcriptional activity by acting as repressors at E-boxes (*Yang and Hurlin, 2017*). In principle, either activation of MYC or inactivation of MXD function would stimulate expression of overlapping gene targets and potentially promote oncogenesis. Some support for this notion is provided by recent TCGA data, indicating that a subfraction of multiple tumor types are subject to deletions of MXD family proteins (*Cancer Genome Atlas Network et al., 2018*). However, most striking are the high frequency of genetic alterations sustained by the MAX dimerizing repressor, MGA (pronounced mega), across a wide range of cancers including 8% of lung adenocarcinomas (*Cancer Genome Atlas Research Network, 2014*; *De Paoli et al., 2013*; *Cancer Genome Atlas Network et al., 2018*; *Sun et al., 2017*). The majority of MGA mutations in lung adenocarcinoma are truncating mutations (*Cancer Genome Atlas Research Network, 2014*), and enrichment in nonsense mutations at the MGA locus across all cancers is statistically significant, suggestive of a tumor suppressive role for the protein (*ICGC Breast Cancer Group et al., 2014*). Consistent with this, MGA emerged as one of the top hits in a genome-wide CRISPR screen for tumor suppressors in DLBCL (diffuse large B-cell lymphoma) lines (*Reddy et al., 2017*).

We identified and initially characterized MGA as a factor that specifically binds the small bHLHZ protein MAX (*Hurlin et al., 1999*). MAX is the obligate dimerization partner of the MYC family of oncogenic drivers (*Amati et al., 1993*; *Blackwood and Eisenman, 1991*). MGA is unique in that it is a dual specificity transcription factor that contains both a TBX domain and a bHLHZ region (*Hurlin et al., 1999*). However, MGA is among the least studied members of the network. Studies in mice and zebrafish suggest that MGA plays an important role in early development (*Rikin and Evans, 2010*; *Sun et al., 2017*; *Washkowitz et al., 2015*). Specifically, loss of MGA leads to peri-implantation lethality due to defects in polyamine biosynthesis (*Washkowitz et al., 2015*). Much of what is known of MGA's function in mammalian cells results from studies in mouse embryonic stem cells (mESCs). An RNAi screen revealed that MGA and its dimerization partner MAX are both involved in repressing germ cell-related transcripts (*Maeda et al., 2013*) and act to prevent entry into meiosis in germline stem cells by directly repressing meiotic genes (*Suzuki et al., 2016*). Moreover, a number of studies have shown that MGA interacts with non-canonical PRC1.6 complexes (herein ncPRC1.6) in ES cells and plays an integral role in recruiting ncPRC1.6 members to target promoters (*Fursova et al., 2019*; *Gao et al., 2012*; *Ogawa et al., 2002*; *Scelfo et al., 2019*; *Stielow et al., 2018*). Despite these advances in understanding MGA function in a developmental context, MGA's role in tumorigenesis remains largely uncharacterized.

A recent study revealed that ectopic expression of MGA in lung adenocarcinoma cell lines retards their growth (*Llabata et al., 2020*). Here, we demonstrate that MGA loss of function promotes tumorigenesis in vivo and investigate the molecular basis of its tumor suppressive activity. Given that MGA activity appears to be compromised at high frequency in lung adenocarcinoma, we utilized a lentiviral based in vivo CRISPR strategy to inactivate *Mga* in murine lung cancer models coupled with in vitro functional and genomic occupancy studies to elucidate the functional and molecular consequences of MGA loss. MGA is comprised of over 3000 amino acids and is the largest of the proteins within the MYC network. In addition to its N-terminal T-Box binding motif and C-terminally localized bHLHZ domain which interacts with MAX, MGA possesses a conserved segment of unidentified function (DUF4801) (*Blum et al., 2021*) as well as extended amino acid sequences whose functional role remains a mystery. In this paper, we extend our murine model of MGA as a lung tumor suppressor by focusing on the bHLHZ region and the DUF domain to begin to characterize the role of the MGA-MAX heterodimer and its associated ncPRC1.6 complex factors in driving transcriptional repression in malignant settings. Lastly, we examined the potentially broader function of MGA as a tumor suppressor by studying the functional consequences of MGA loss in normal and malignant colorectal organoids.

## Results

### *Mga* inactivation in *Kras*^G12D^ and *Kras*^G12D^*Trp53*^−/−^-driven mouse models of lung cancer leads to accelerated tumorigenesis

Publicly available datasets had indicated a correlation between deletion or truncating mutations of several members of the proximal MYC network and tumor progression in lung cancer. Focusing on MGA, we note that it is mutated or deleted in 53 of 507 (10%) lung adenocarcinoma patients (TCGA) and in 233 of 3696 (6%) patients sequenced through the GENIE project (*Figure 1—figure supplement 1A*). The majority of MGA mutations were truncating with approximately 25% homozygosity in the TCGA LUAD dataset, suggestive of selection for inactivating mutations in tumorigenesis. In addition, low levels of MGA in lung adenocarcinoma significantly correlate with decreased overall survival (*Figure 1—figure supplement 1B*; *Győrffy et al., 2013*). To ascertain whether *Mga* functions as a tumor suppressor in vivo, we utilized CRISPR-CAS technology to inactivate *Mga* in a *Kras*^LSL-G12D/+^ (henceforth termed Kras) driven mouse model of lung cancer (*Jackson et al., 2001*; *Sánchez-Rivera et al., 2014*). Intratracheal lentiviral delivery of Cre, Cas9, and sgRNA against *Mga* using a lentiCRISPRv2 Cre vector (*Walter et al., 2017*; *Figure 1A*) results in the activation of *Kras*^G12D^ along with indel formation at the *Mga* locus (*Figure 1—figure supplement 1C*). For survival analysis, mice were euthanized when moribund from lung tumor burden. There was a significant decrease in survival in *Kras*^G12D^ mice receiving sgMga when compared to mice receiving the lentiCRISPRv2 vector containing Cre and Cas9 but lacking sgMga (referred to herein as 'empty' vector; *Figure 1B*). Tumors from sgMga-treated mice exhibited a higher proliferation index as ascertained by Ki67 staining when compared to mice infected with empty vector controls (*Figure 1C,D*). Pathological examination of lesions at endpoint by a board certified veterinary pathologist (A.K.) revealed that mice in both groups had a spectrum of grade 1–5 tumors characteristic of lesions that develop in the Kras model (*Figure 1—figure supplement 1D,E*; *Jackson et al., 2005*). Sequencing of tumors confirmed the presence of indels predicted to result in premature truncation in 75% (6/8) tumors isolated from sgMga-treated mice. We next examined the potential synergy of *Mga* loss with the inactivation of other tumor suppressors. Intriguingly, TCGA and GENIE data revealed that there is a significant co-occurrence of genomic alterations in *MGA* and *TP53* in lung adenocarcinoma patients (*Figure 1—figure supplement 1F*). To further investigate putative cooperation between *MGA* and *TP53 loss*, we utilized the *Kras*^LSL-G12D/+^*Trp53*^fl/fl^ (henceforth termed KP) mouse model of lung cancer. We inactivated *Mga* in this model using the same lentiCRISPRv2 Cre approach described above and analyzed mice at a common time point 3 months later. At 3 months after intratracheal instillation of lentiCRISPRv2cre, KP-sgMga mice had a substantially higher tumor burden when compared to KP mice similarly treated with control vector (*Figure 1E,F*). This was accompanied by an overall increase in number of tumors and tumor grade upon Mga inactivation (*Figure 1G,H*). Immunostaining revealed that sgMga tumors had an increased proportion of Ki67+ tumor cells when compared to controls (*Figure 1I,J*). Consistent with increase in tumor grade, sgMga tumors had increased vasculature (as evidenced by MECA32 staining) when compared to controls (*Figure 1—figure supplement 1G,H*). In addition to analyzing mice at a common 3 month time point, we also aged a separate small cohort of infected animals to isolate tumors and generate cell lines. These cell lines (herein designated KP cells) were utilized to further characterize sgMga-inactivated tumors. A substantial decrease in MGA protein levels was observed in most KP-sgMga cell lines (*Figure 1K,L*, *Figure 1—figure supplement 1I*). We confirmed that KP-sgMga lines are dependent on the inactivation of *Mga* as ectopic expression of MGA in KP-sgMga lines impaired proliferation (*Figure 1M,N*).

### MGA loss de-represses ncPRC1.6 and MYC target genes and upregulates pro-invasion genes

To identify transcriptional changes that may underly the observed tumor phenotypes, we performed RNA-seq analyses. We compared gene expression profiles between 6 Kras-sgMga and 8 Kras lung tumors derived from empty vector control-treated mice (*Figure 2—figure supplement 1A*). sgMga tumors were sequenced and only those confirmed to have indels were used for RNA-seq experiments. Principal component analysis suggested heterogeneity in gene expression across tumors (*Figure 2—figure supplement 1*). This may be due at least in part to varying levels of non-tumor cells such as immune cells and fibroblasts present within tumors in this model (*Busch et al., 2016*).

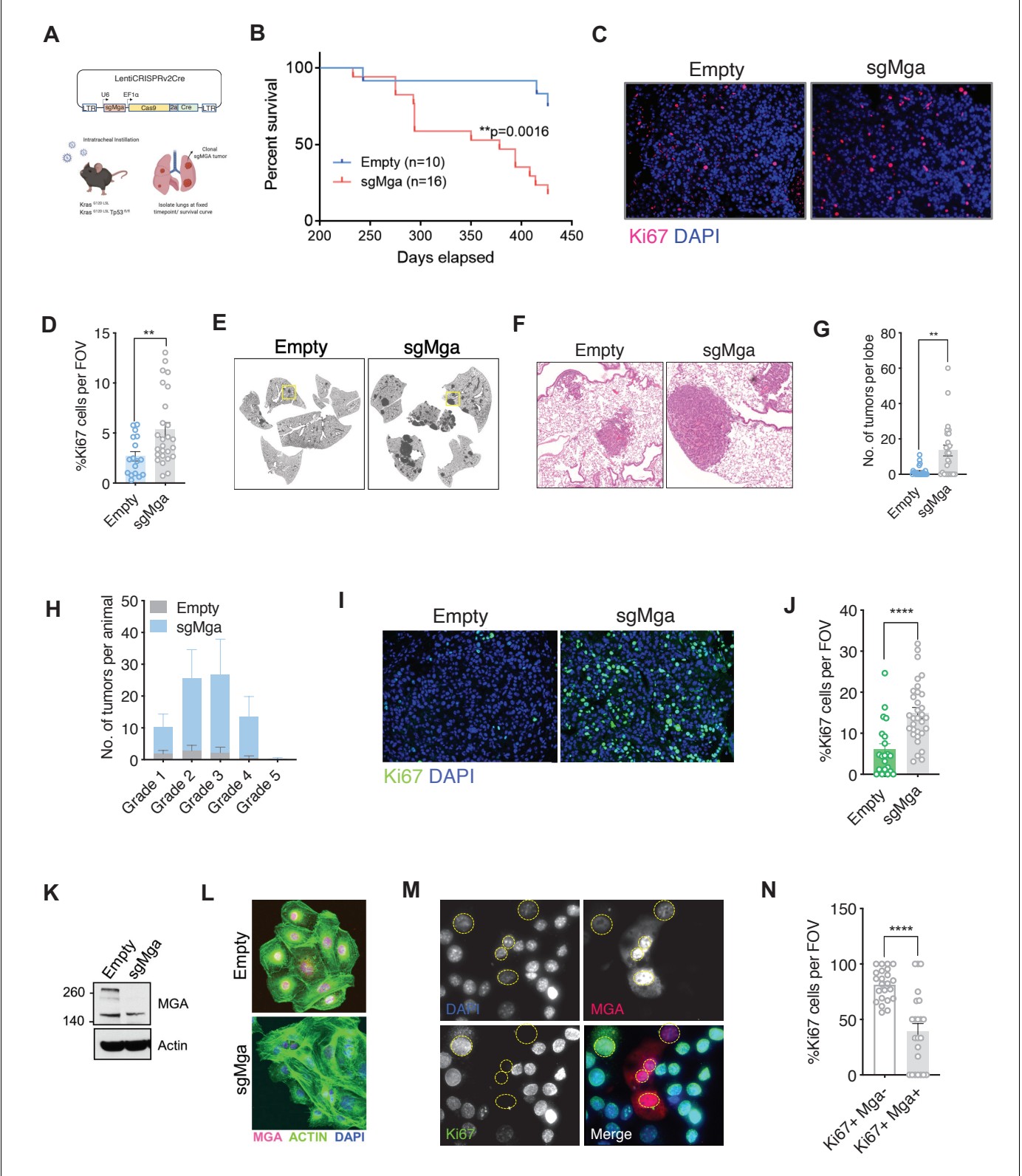

**Figure 1.** *Mga* inactivation in vivo leads to accelerated tumorigenesis. (**A**) Schematic of lentiCRISPRv2cre vector and strategy used for in vivo Kras (K) and Kras, Tp53 (KP) experiments. (**B**) Kaplan–Meier curve for survival in sgMga (Cre-Cas9) vs Empty (Cre-Cas9 lacking sgMga) virus-treated Kras mice (p-value computed using Log-rank test). (**C**) Representative micrographs and (**D**) quantification of Ki67 immunostaining on lung tumor tissue from Kras-sgMga and Kras empty vector infected control mice (n = 5 mice per group. **p=0.0036. Calculated using Welch's t-test assuming unequal variance). (**E**)
*Figure 1 continued on next page*

*Figure 1 continued*

Representative H and E staining of sgMga and Empty lungs from KP mice harvested 3 months post-intratracheal instillation. (F) Higher magnification of selected regions from (E) showing individual tumors in both groups. (G) Quantification of number of lesions per lobe and (H) number of lesions per grade in Empty (control) and sgMga vector treated animals (n = 5 lung lobes each from five mice per group. **p=0.001). (I) Representative images and (J) quantification of Ki67 staining to assess proliferation in KP Empty (control) and KP-sgMga lung tumors harvested 3 months post-infection (n = 32 tumors from four sgMga mice n = 21 tumors from two Empty (control) mice. ****p<0.0001. Calculated using Welch's t-test assuming unequal variance). (K) Western blot and (L) immunofluorescent staining for MGA in cell lines derived from KP tumors. (M) Representative images and (N) quantification of Ki67 and MGA co-staining in an sgMga KP line upon ectopic expression of MGA (n = 23 FOV each from three independent experiments. ****p<0.0001. All p-values calculated using a two-sided Student's t-test unless otherwise noted. Error bars represent SEM.

The online version of this article includes the following figure supplement(s) for figure 1:

**Figure supplement 1.** *Mga* inactivation in vivo leads to accelerated tumorigenesis.

Despite the heterogeneity, Mga-inactivated cells exhibited a striking increase in expression of several meiotic genes, such as *Stag3*, *Sox30*, and *Tdrd1*, previously reported to be directly regulated by MGA-PRC1.6 in ES and germline cells (*Figure 2A*, *Supplementary file 1a*; *Suzuki et al., 2016*). This suggests that, similar to ES cells, MGA also assembles into ncPRC1.6 complexes in somatic normal and tumor cells (*Figure 2A*; *Llabata et al., 2020*). Consistent with our Ki67 data, *Ccnd1* levels were elevated in *sgMga* tumors compared to control (*Figure 2A*, *Supplementary file 1a*). Gene set enrichment analysis revealed that several metabolic pathways and a subset of MYC-regulated genes were also upregulated in sgMga tumors, indicating that MGA represses MYC targets, similar to repression by MXDs seen in other tumor models (*Figure 2B*; *Yang and Hurlin, 2017*). However, inactivation of *Mga* led to the downregulation of genes involved in anti-tumor responses, such as NK cell markers and interferon signaling genes (*Figure 2C*, *Figure 2—figure supplement 1C*). Because MYC has been reported to repress anti-tumor immune responses in tumors, including in Kras-driven lung and pancreatic adenocarcinomas (*Casey et al., 2016*; *Kortlever et al., 2017*; *Muthalagu et al., 2020*), MGA may normally act to limit MYC repression of these genes.

Next, we performed RNA-Seq on MGA-inactivated and control KP lines. Cell lines were used to overcome the limitation of having a mixed population of cells in primary tumors as with the Kras tumors. Despite the differences in genetic background (Kras vs. KP), several ncPRC1.6 complex repressed germ cell related transcripts, including *Stag3*, were highly upregulated in KP cells lacking MGA (*Figure 2D*, *Supplementary file 1b*), similar to Kras-sgMga tumors. In addition, we also observed an upregulation of pro-invasive genes such as *Podxl2*, *Gpc2*, *Snai2*, and *Itga1* (*Figure 2D*). *Gpc2* and *Podxl2* also appeared to be increased in Kras-sgMga tumors (*Figure 2A*). Gene set enrichment analysis revealed that epithelial to mesenchymal transition (EMT) and TGF-beta signaling were amongst the most significantly enriched pathways in sgMga KP cells (*Figure 2E*). Intriguingly, genes upregulated in sgMga KP cells correlated significantly with those upregulated upon MYC, E2F4, and TWIST1 overexpression in other systems (*Figure 2—figure supplement 1D*). In addition, there was also a significant correlation with genes reported to be upregulated upon PCGF6 knockdown or MYC overexpression (*Figure 2—figure supplement 1D*). Along with the observed spectrum in tumor grades and MECA32 staining in the KP model, our results corroborate a role for MGA in repressing invasion (*Figure 1—figure supplement 1D,E, and G*, and see below). We noticed some commonalties in pathways de-repressed upon MGA loss in Kras and KP tumors. Processes such as hypoxia, glycolysis, carbohydrate, and lipid metabolism, found to be enriched in sgMga Kras tumors, also scored significantly in the KP cells (*Figure 2B,E*). We employed qPCR to further confirm the activation of EMT and pro-invasive genes like *Snai1*, *Itga1*, *Podxl2*, and *Gpc2* (*Figure 2F*) and ncPRC 1.6 targets *Stag3* and *Tdrd1* (*Figure 2G*) in a panel of control and sgMga KP lines. However, as determined by gene set enrichment, no significant overlap amongst genes or pathways downregulated in KRas vs KP-sgMga cells was evident (*Figure 2—figure supplement 1E*).

Taken together, our data suggest that MGA regulates meiotic transcripts such as *Stag3* as well as multiple MYC targets in lung tumor initiating cells that overlap with previously reported MGA targets in ES cells as well as distinct gene signatures based on cellular context, such as *Trp53* expression.

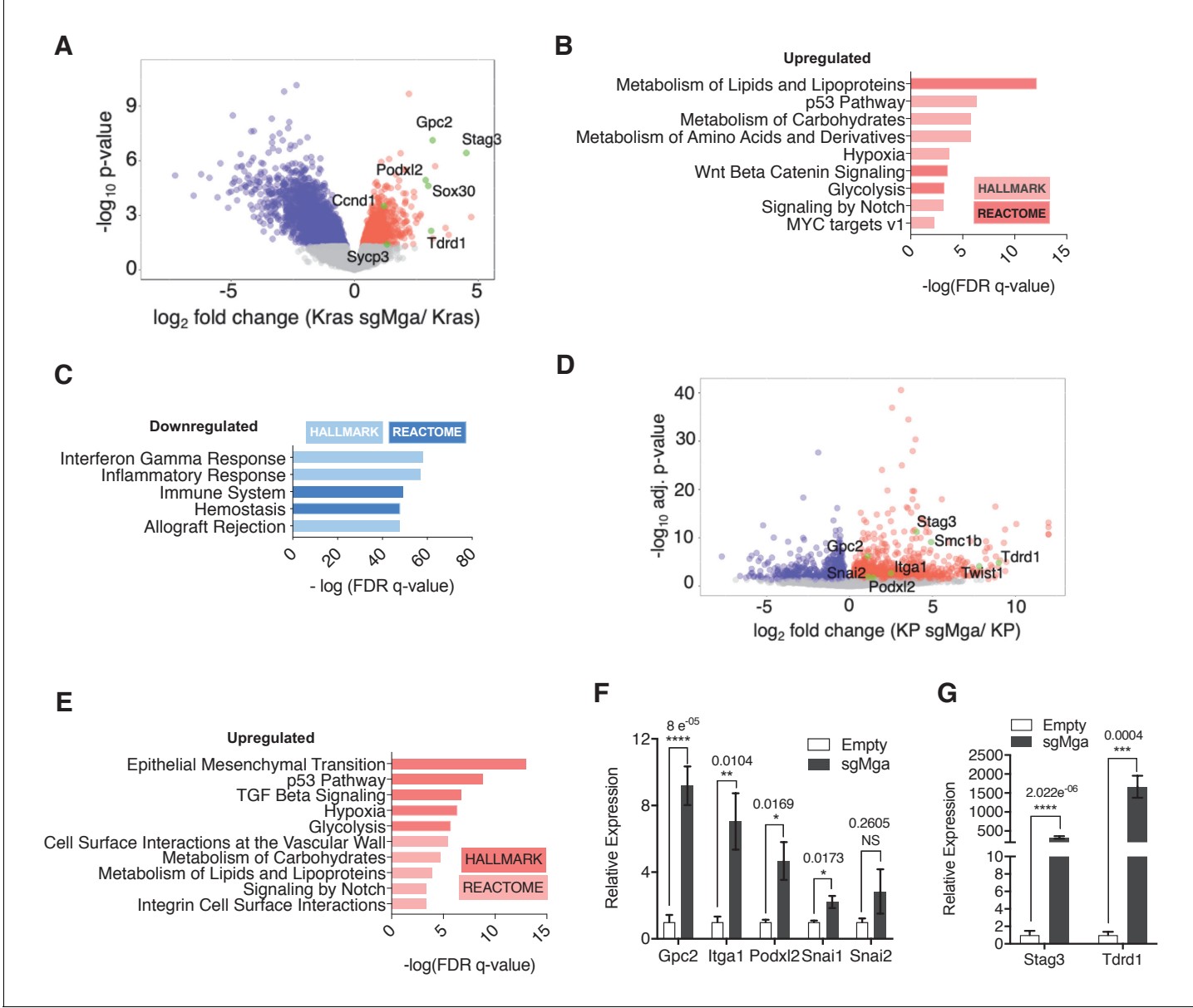

**Figure 2.** MGA loss leads to the de-repression of PRC1.6 and MYC targets and upregulation of pro-invasion genes. (**A**) Volcano plot of Kras tumor RNA profiling data comparing Kras and Kras-sgMga tumors (see *Figure 1B–D*). Upregulated pro-invasion and PRC1.6 targets highlighted in green. (**B, C**) Hallmark and Reactome GSEA for pathways enriched for in (**B**) upregulated and (**C**) downregulated genes in Kras-sgMga tumors compared to Kras empty control tumors. (**D**) Volcano plot of KP cell line RNA-Seq data showing PRC1.6 and pro-invasive genes in green (n = 2 cell lines in triplicate for KP control and 3 cell lines in triplicate for KP-sgMGA). (**E**) Hallmark and Reactome analysis of genes upregulated in KP-sgMga vs. control KP empty cell lines. (**F, G**) qPCR on Empty and sgMga KP lines to confirm levels of (**F**) pro-invasive and (**G**) PRC1.6 meiotic targets (n = 6 for each – 2 sets of RNA [biological replicates] from three different lines for sgMga and Empty). p-values for individual genes indicated in figure. All p-values calculated using a two-sided Student's t-test. Error bars represent SEM.

The online version of this article includes the following figure supplement(s) for figure 2:

**Figure supplement 1.** MGA loss leads to the de-repression of PRC1.6 and MYC targets and upregulation of pro-invasion genes.

## MGA associates with ncPRC1.6 subunits through its DUF region and stabilizes the complex

Given that MGA inactivation is associated with the consistent upregulation of similar genes and pathways in Kras and KP tumor cells (*Figure 2*) as well as in human lung cancer lines (see below), we

sought to identify proteins that interact with MGA and mediate its repressive activity. We performed tandem affinity purification of double tagged (FLAG and HIS) MGA in 293 T cells followed by mass spectrometry to identify MGA interactors. We confirmed that full-length MGA or MGA fragments (aa967–1300 and aa2153–2856) interact with several repressive molecules and complexes (protein prophet score > 0.9). These include members of the ncPRC1.6 complex such as RING2 and L3MBTL2 (*Figure 3A*). In addition, MGA also associated with WDR5, RBBP5, ASH2L, and DPY30 (members of the chromatin modifier MLL-WRAD complex) and HDACs 1 and 2 (*Figure 3A*). Previous studies in ES cells indicated that MGA is required for ncPRC1.6 assembly and stabilizes some components of the complex (*Gao et al., 2012*; *Scelfo et al., 2019*; *Stielow et al., 2018*). Therefore, we determined levels of PRC1.6 members MAX, L3MBTL2, and E2F6 in *Mga*-inactivated and control KP lines. Mga loss resulted in decreased protein levels of E2F6, PCGF6, and L3MBTL2 (*Figure 3B–E*) despite equivalent mRNA levels (*Figure 3—figure supplement 1*), whereas MAX levels remain unchanged (*Figure 3B*, *Figure 3—figure supplement 1*). We further confirmed decreased E2F6, L3MBTL2, and PCGF6 protein levels in *MGA* mutant human non-small cell lung cancer lines H2291 and Lou-NH-91 when compared to wild-type lines such as NCI-H1975, NCI-H23, and 91T (*Figure 3F–I*). Our data indicate that MGA functions in maintaining the integrity of the PRC1.6 complex even in malignant cells.

We next identified the region within MGA that mediates its putative scaffolding function for PRC1.6. Our initial mass spectrometric analysis of immunoprecipitates using fragments of MGA in 293FT cells revealed that the region encompassing aa 967–1300 interacts with L3MBTL2 (an obligate component of PRC1.6) and several other proteins (*Figure 3A*). Interrogation of protein domain databases revealed that this region includes a conserved domain of unknown function (DUF4801, located at aa 1043–1084). To assess whether this region in MGA is required for binding to L3MBTL2 in vitro we performed co-immunoprecipitation experiments using HA-tagged L3MBTL2 and Flag-tagged MGA or MGAΔDUF (MGAΔ1003–1304 deleting a segment encompassing DUF 4801). *Figure 4A* shows preferential association of L3MBTL2 with wild-type MGA as compared with MGAΔDUF. Analysis of publicly available TCGA Pancancer data reveals multiple mutations across MGA. Focusing on lung adenocarcinomas, TCGA sequencing data revealed that multiple truncating mutations in MGA occur within the AA1003–1304 region surrounding DUF4801 (*Figure 4—figure supplement 1A,B*). To study the functional importance of this region, we ectopically expressed either the WT or MGAΔDUF versions in two different sgMga mouse KP lines and observed that addback of MGAΔDUF had no effect on proliferation of KP cells whereas expression of WT MGA led to a significant decrease in proliferation (*Figure 4B–D*). We then transduced the human MGA mutant lung squamous cell line LOU-NH-91 with full-length MGA or MGAΔDUF and observed an increase in L3MBTL2 protein levels upon wild type MGA addition but not upon MGAΔDUF expression (*Figure 4E*), although both forms are capable of nuclear localization (*Figure 4—figure supplement 1C*). MYC levels and, to a lesser extent, MAX levels decreased in MGA and MGAΔDUF over-expressing cells, suggesting feedback regulation within the network (*Figure 4E*). In contrast to full-length MGA addback, which resulted in a drastic reduction in cell number, MGAΔDUF expression led to only a mild impairment in proliferation suggesting that the association of ncPRC1.6 is unlikely to account for all of MGA's antiproliferative activity (*Figure 4F*).

## MGA is a determinant of PRC1.6 genomic binding in tumor cells

Genomic targets of the PRC1.6 complex have been identified in several cellular contexts, including in 293FT cells and mouse ES and germline cells (*Endoh et al., 2017*; *Gao et al., 2012*; *Scelfo et al., 2019*; *Stielow et al., 2018*; *Suzuki et al., 2016*). To assess the direct role of ncPRC1.6 complexes in tumor cells, we performed conventional ChIP-Seq (X-ChIP: crosslinked chromatin immunoprecipitation followed by sequencing). We determined MAX, MGA, MYC, L3MBTL2, E2F6, and RNA polymerase II (RNA pol2, all phosphorylated forms) occupancy in *Mga*-inactivated and control (empty) KP tumor-derived cells lines. MGA, MAX, E2F6, and L3MBTL2 were seen to bind several thousand promoters in MGA-expressing KP tumor cells suggesting that the PRC1.6 complex plays a broad role in tumor cell gene expression (*Supplementary file 1c*). Strikingly, 85% of MAX, and nearly all L3MBTL2 and E2F6 binding loci, are also occupied by MGA (*Figure 5A*). Moreover, we noted a marked reduction in L3MBTL2 and E2F6 binding in KP-sgMga cells, while overall MAX and MYC genomic occupancy was only modestly affected (*Figure 5B,C*). These binding data are consistent with a role for

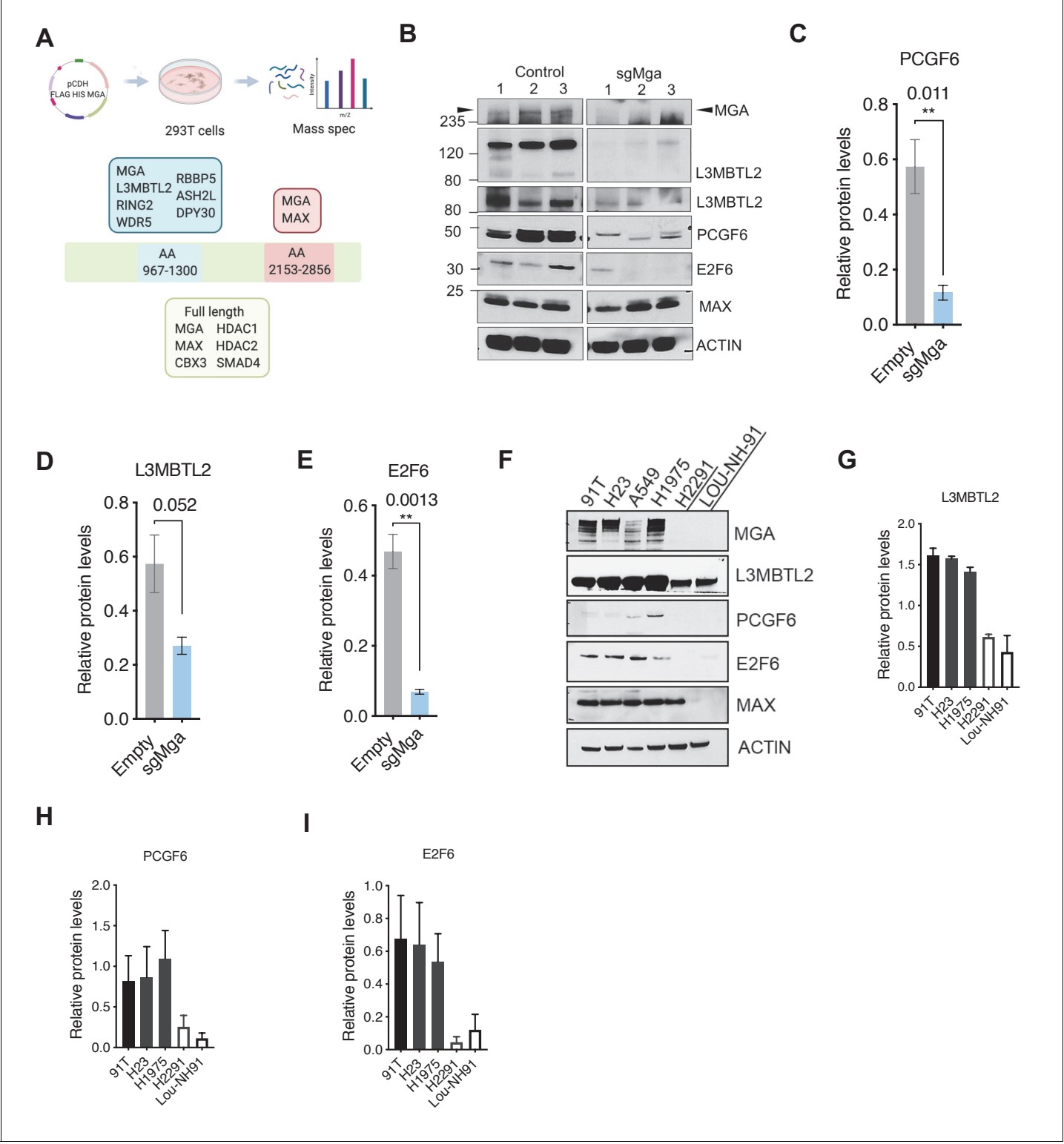

**Figure 3.** MGA binds and stabilizes PRC1.6 complex members in lung cancer cells. (**A**) Mass spectrometric analysis of proteins interacting with FLAG-His-double tagged full-length MGA or isolated aa967–1300 (encompassing the DUF4801 region) and aa2153–2853 (encompassing the bHLHZ domain) segments of MGA. Confirmed interactors shown shaded in color (protein prophet score > 0.9). (**B**) Representative immunoblot to show protein levels of MGA, L3MBTL2 (lower panel to show L3MBTL2 at higher exposure), PCGF6, E2F6, and MAX in control and sgMga mouse KP lines (n = 3 individual cell lines for control and sgMga). (**C–E**) Densitometry of indicated PRC1.6 members, in control and sgMGA KP lines. (**F**) Representative immunoblots for the indicated proteins in MGA WT and mutant human lung adenocarcinoma cell lines. (**G–I**) Quantification of protein levels of the indicated proteins in

*Figure 3 continued on next page*

*Figure 3 continued*

MGA WT (blacks bars) vs. MGA mutant (white bars) lung cancer lines. For densitometry, three independent western runs were used for each comparison. Error bars represent SEM.

The online version of this article includes the following figure supplement(s) for figure 3:

**Figure supplement 1.** PRC1.6 complex member mRNA levels unchanged upon MGA loss in lung cancer cells.

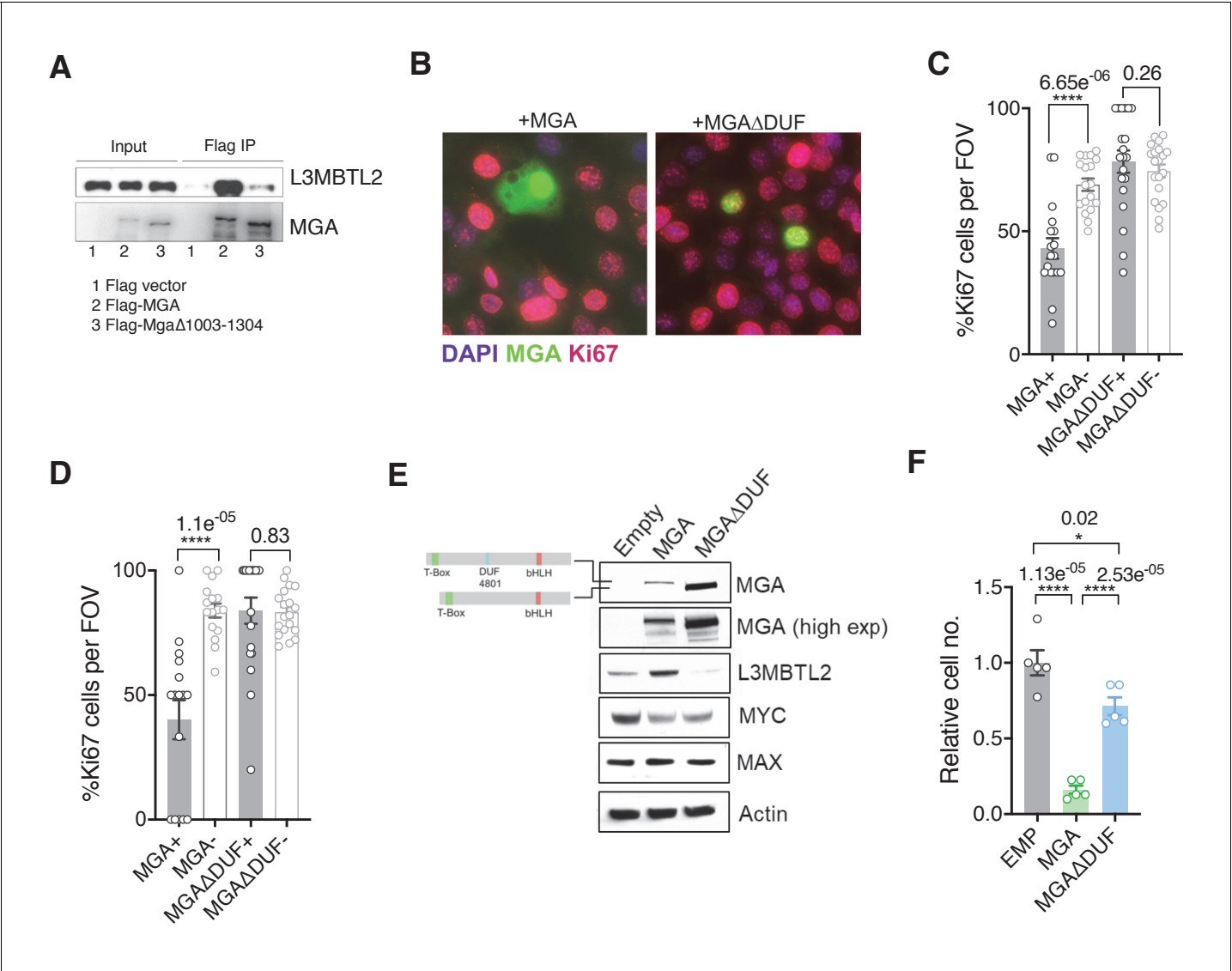

**Figure 4.** The region containing the DUF4801 domain is critical for MGA's tumor suppressive function. (**A**) Immunoblot showing co-immunoprecipitation of L3MBTL2 with full length or MGAΔ1003–1304 (ΔDUF). (**B**) Representative co-staining for Ki67 and MGA deficient mouse KP cells expressing MGA or MGAΔ1003–1304 (ΔDUF). (**C, D**) Quantification of Ki67+ cells in two independent KP-sgMga lines: (**C**) H8712 T3 (n = 18 FOV MGA, n = 20 FOV MGAΔDUF from four biological replicates each) and (**D**) H8638 (n = 17 FOV MGA, n = 20 FOV MGAΔDUF from at least four biological replicates each). (**E**) Immunoblots of ectopic expression of MGA or MGAΔDUF in LOU-NH-91 human lung squamous carcinoma cell line. (**F**) Relative growth of MGA and MGAΔDUF expressing LOU-NH-91 cells (n = 5 replicates across three independent experiments for each group). All p-values calculated using a two-sided Student's t-test unless otherwise noted (*p<0.05, ****p<0.0001). Error bars represent SEM.

The online version of this article includes the following figure supplement(s) for figure 4:

**Figure supplement 1.** Cancer associated mutations within MGA.

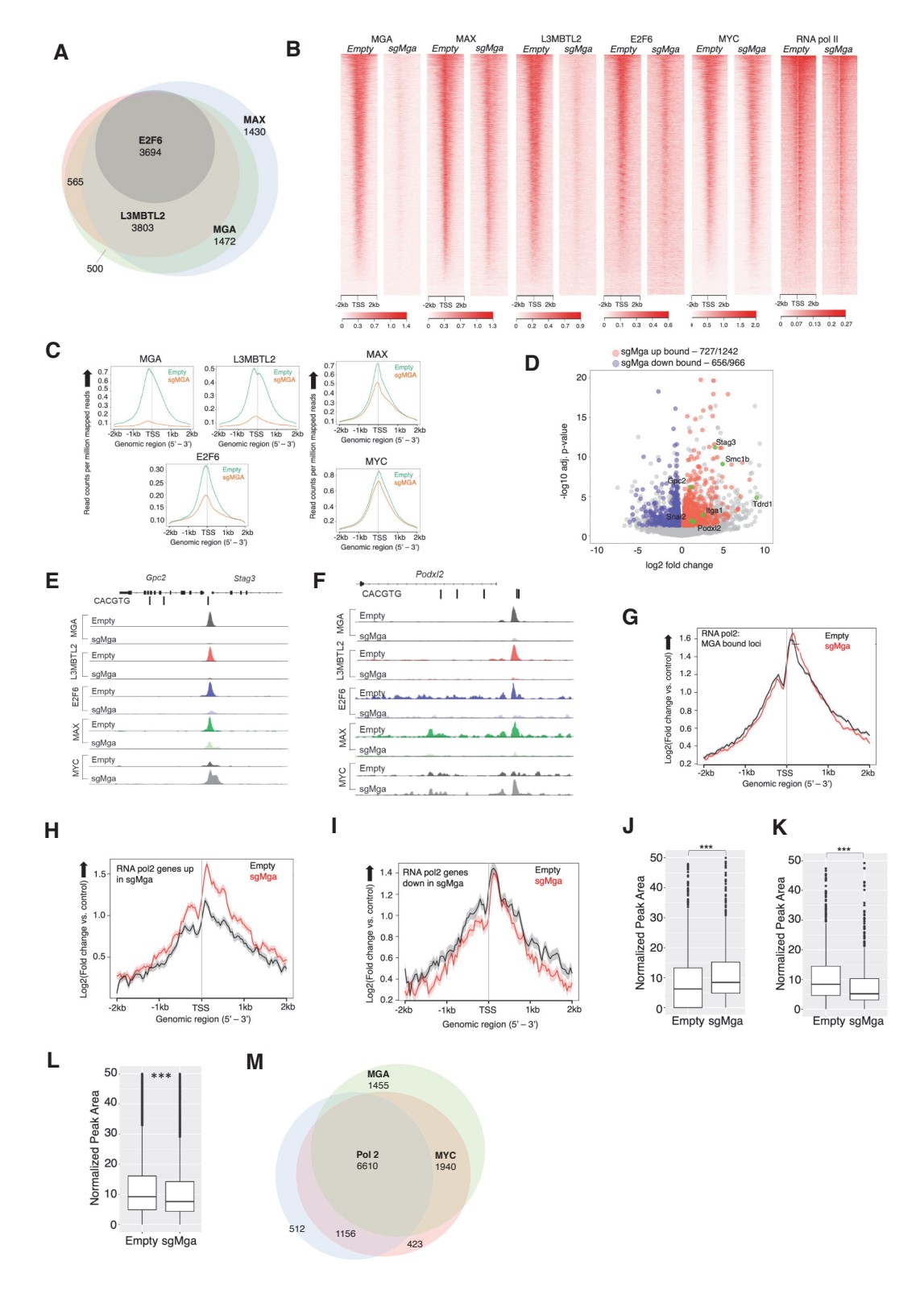

**Figure 5.** MGA is essential for PRC1.6 genomic binding in tumor cells. (**A**) Venn diagram showing overlap between MAX, MGA, E2F6, and L3MBTL2 bound genes in KP cells. (**B**) Heatmaps showing genome-wide promoter proximal (±2 kb) binding by the indicated transcription factors in control and sgMga KP cells. (**C**) Meta-plots of occupancy by the indicated transcription factors in Empty and sgMga KP cells. (**D**) Volcano plot of differentially expressed genes that are directly bound by MGA (red and blue dots indicate bound genes, up and down as indicated; green dots with labels indicate

*Figure 5 continued on next page*

Figure 5 continued

genes functionally implicated in MGA activity). (E, F) representative tracks for MGA, L3MBTL2, E2F6, MAX, and MYC binding at the (E) Stag3/Gpc2 promoter and (F) Podxl2 loci in KP cells and sgMGA KP cells. (G–I) Meta-plots of RNA pol2 enrichment at the TSS ±2 kb in sgMGA KP cells (red lines) vs KP cells with wild-type MGA (black lines) at (G) MGA-bound genes, (H) genes upregulated, and (I) genes downregulated in sgMGA KP cells. (J–L) Box plots of RNA pol2 peak areas at TSS ±2 kb in KP vs sgMGA KP cells for (J) loci upregulated (p-value=2.1e-07), (K) loci downregulated (p=1.6e-06). The p-values in (J–L) calculated by Welch two sample t-test. (L) Box plots of RNA pol2 peak areas at MYC-bound loci in KP and sgMGA-KP cells (p=8.7e-05). (M) Venn diagram depicting the numbers of genes bound by RNA polymerase II, MYC, and MGA and their extent of overlap in KP cells.

The online version of this article includes the following figure supplement(s) for figure 5:

**Figure supplement 1.** MGA, PRC1.6, and MYC binding and activity in human and murine LUAD-derived cells.

MGA-MAX in both recruitment and stabilization of PRC1.6 complex members (*Figures 3B*, *4E*, and *5A*).

Regarding altered gene expression, we observe MGA binding to the majority of genes significantly upregulated and downregulated upon MGA loss (*Figure 5D*). Focusing on several key genes previously associated with MGA function, we detected significant enrichment of both MGA and ncPRC1.6 subunits (i.e. L3MBTL2 and E2F6) binding at *Stag3/GPC2* and *Podxl2* promoters, which are upregulated in sgMga-KP cells (*Figure 5E,F*). Of note, MYC and MAX binding at promoters in MGA null cells is largely unperturbed and even increased in certain cases (*Figure 5E,F*). For example, MYC binding is augmented at *Stag3* and *Podxl2* in sgMga cells, in contrast to decreased occupancy of L3MBTL2 and E2F6 at the same loci (*Figure 5E,F*, *Figure 5—figure supplement 1B,C*).

To further explore the effects of MGA loss on gene expression, we assessed RNA polymerase II occupancy at MGA-bound loci (*Figure 5G–I*). We detect nearly equivalent amounts of RNA polymerase II associated with the TSS of all loci that were targeted by MGA in both the MGA-expressing and MGA-deleted KP cells (*Figure 5G*). However, when we only bin genes whose expression is increased upon MGA loss, we observe higher levels of TSS-proximal RNA polymerase II in the MGA-deleted cells relative to the control cells with wild-type levels of MGA (*Figure 5H*). By contrast, genes whose expression is downregulated upon MGA deletion exhibit nearly the same relative levels of RNA polymerase II binding to the TSS, although the regions flanking the TSS show somewhat lower levels of polymerase in the sgMGA cells (*Figure 5I*). This idea is supported by comparison of box plots of RNA polymerase II peak areas ± 2 KB for the sets of upregulated (*Figure 5J*) and down-regulated (*Figure 5K*) genes.

To determine whether the above genomic findings apply to human lung cancer cells, we transduced the MGA-expressing A549 human lung cancer line with a lentiCRISPRv2 vector containing Cre and Cas9 either lacking (empty control) or containing sgMGA. We probed for MAX, E2F6, L3MBTL2, and MYC occupancy using CUT and RUN in the sgMGA and control lines (*Janssens et al., 2018*). As for mouse KP lines, we observed a overall decrease in MGA and L3MBTL2 binding in *MGA*-inactivated cells (termed sgMGA), including at STAG3 and *PODXL2* promoters (*Figure 5—figure supplement 1A*; *Supplementary file 1d*). Concomitant with an increase in STAG3 and PODXL2 expression in sgMGA A549 cells, there was decreased occupancy of MGA and L3MBTL2 at their respective promoters accompanied by an increase in MYC binding (*Figure 5—figure supplement 1B,C*). Taken together, this data strongly suggests that MGA mediates ncPRC1.6 binding to MYC and E2F targets in human LUAD tumor cells. Therefore, loss of MGA abrogates PRC1.6 occupancy at several thousand promoters. MYC-MAX binding is largely unchanged at these promoters but increases within a specific subset.

## Involvement of MYC in the MGA deletion phenotype

Given that MGA-MAX heterodimers can arguably be considered as members of the MYC network, we assessed the contribution of MYC-MAX heterodimers to the transcriptional and phenotypic consequences of MGA loss of function. First, we examined TCGA data to interrogate co-occurrence of MGA and MYC paralog alterations in a panel of LUAD patient samples. Analysis of this data indicated that amplification of MYC paralogs is rare in LUAD whether or not MGA is inactivated (e.g. 1/23 cases of MGA truncating or splice site mutations exhibited amplification of MYCL, while 7/460 cases with wild-type MGA had amplification of any MYC paralogs). Moreover, TCGA samples did not indicate significantly altered MYC family RNA levels in MGA mutant tumors compared with wild-

type MGA tumors (*Figure 5—figure supplement 1D*). This is consistent with the very modest change in genome-wide MYC association with TSSs in sgMGA-KP cells vs. controls (*Figure 5B,C*). We also examined RNA pol2 association with MYC binding sites in sgMGA vs wild-type cells. The peak areas plotted in *Figure 5M* suggest an overall decrease in RNA polymerase II association with MYC-bound genes upon MGA loss despite the fact that MYC binding increases at a subset of loci in MGA deleted cells (examples shown in *Figure 5E,F*, *Figure 5—figure supplement 1B,C*).

To ask whether the endogenously expressed MYC is critical for growth, we deleted *MYC* in sgMGA-KP and control KP tumor lines and found suppression of proliferation in both cases (*Figure 5—figure supplement 1E*). In addition, treatment of control and sgMGA A549 cells with 10058-F4, an extensively used MYC-MAX dimerization inhibitor (*Yin et al., 2003*) revealed that MGA deficient cells are as sensitive to MYC inhibition as control vector-transduced cells (*Figure 5—figure supplement 1F*). A similar trend is observed for A549 cells upon treatment with MS2-008 (*Struntz et al., 2019*), a chemical probe that inhibits MYC-MAX-driven transcription (*Figure 5—figure supplement 1G*). Together, these data indicate that MYC levels, although evidently not amplified, are likely to be essential for cell survival growth in both MGA WT and MGA deleted cells.

## ncPRC1.6 subunits contribute to MGA control of gene expression

To begin to examine the role of ncPRC1.6 complex subunits in regulation of gene expression by MGA, we employed Crispr Cas9 to individually inactivate *L3mbtl2*, *Pcgf6*, or *Mga* in the mouse tumor derived KP cells (*Figure 6—figure supplement 1A*). Loss of the cognate encoded proteins had little effect on growth of KP cells, an expected result as KP cells and the knockout cells are both related and oncogenically transformed. Nonetheless, our RNA-Seq analysis revealed significant overlap among genes upregulated upon MGA loss and those upregulated by individual CRISPR inactivation of *L3mbtl2* and *Pcgf6.* Most notably we find that genes associated with meiotic recombination, repair or outgrowth are among the top 50 genes upregulated upon MGA loss as well as by reduction of PCGF6 or L3MBTL2 levels (e.g. *Stag3*, *Tdrkh*, *Zcwpw1*, *Gpc2*) (*Figure 6A,B*). These findings support the notion of a critical role for ncPRC1.6 complex members in gene regulation by MGA. However, the data indicate that for the majority of genes neither single inactivation of *L3mbtl2*, *Pcgf6*, nor CRISPR inactivation of both *L3mbtl2* + *Pcgf6*, lead to upregulation to the same overall extent as the MGA deletion itself. This is also true for genes that are downregulated upon MGA loss (*Figure 6—figure supplement 1B*). The data imply that other ncPRC1.6 subunits (such as E2F6 and CBX3) or other proteins (such as HDAC1 and HDAC2) that are apparently not associated with the MGA DUF4801 region contribute to overall transcriptional effects of MGA (*Figure 3A*).

## MGA loss correlates with activation of *STAG3* and *PODXL2* in human lung cancer and results in a pro-invasive phenotype in vitro

We next asked if the phenotypic and expression changes we observed in our murine lung adenocarcinoma model studies are also manifested in patients and established human lung cancer cell lines. Given the overlap of MGA targets we observed in our mouse models, we examined human lung adenocarcinoma TCGA data to assess whether the expression of these genes correlates with *MGA* loss in patients. We then grouped patients into MGA WT and MGA altered groups, based on the presence of mutations and deletions at the *MGA* locus, to assess functional enrichment of expression of genes in MGA altered tumors. Interestingly, STAG3 appeared to be highly expressed (SD > 3 above mean) in a significant subset of MGA-altered cases (truncating mutations and deletions) (*Figure 7—figure supplement 1A,B*). Genes overexpressed in the MGA-altered group are significantly enriched for MYC, MAX, E2F6, and E2F4 in Enrichr ChEA analysis (*Figure 7—figure supplement 1C*). Consistent with mouse Kras tumor data, there appeared to be a significant downregulation of anti-tumor immune responses and decrease in IFN signaling genes such as *IFITM1* in MGA-altered cases (*Figure 7A,B*, *Figure 7—figure supplement 1D*). We then sought to overlap genes upregulated in our mouse Kras and KP expression profiling data (>1.5× fold change, adj. p-value<0.05) with genes overexpressed in the MGA-altered TCGA group (p-value<0.05). Amongst the five gene overlap, we observed that established ncPRC1.6 target STAG3 and pro-invasion gene PODXL2 were significantly altered in all three comparisons (*Figure 7C–E*).

To investigate whether *MGA* loss leads to similar functional consequences in vitro, we used either shRNA- or CRISPR-based approaches to delete or knock down *MGA* in A549 and NCI-H23 lung

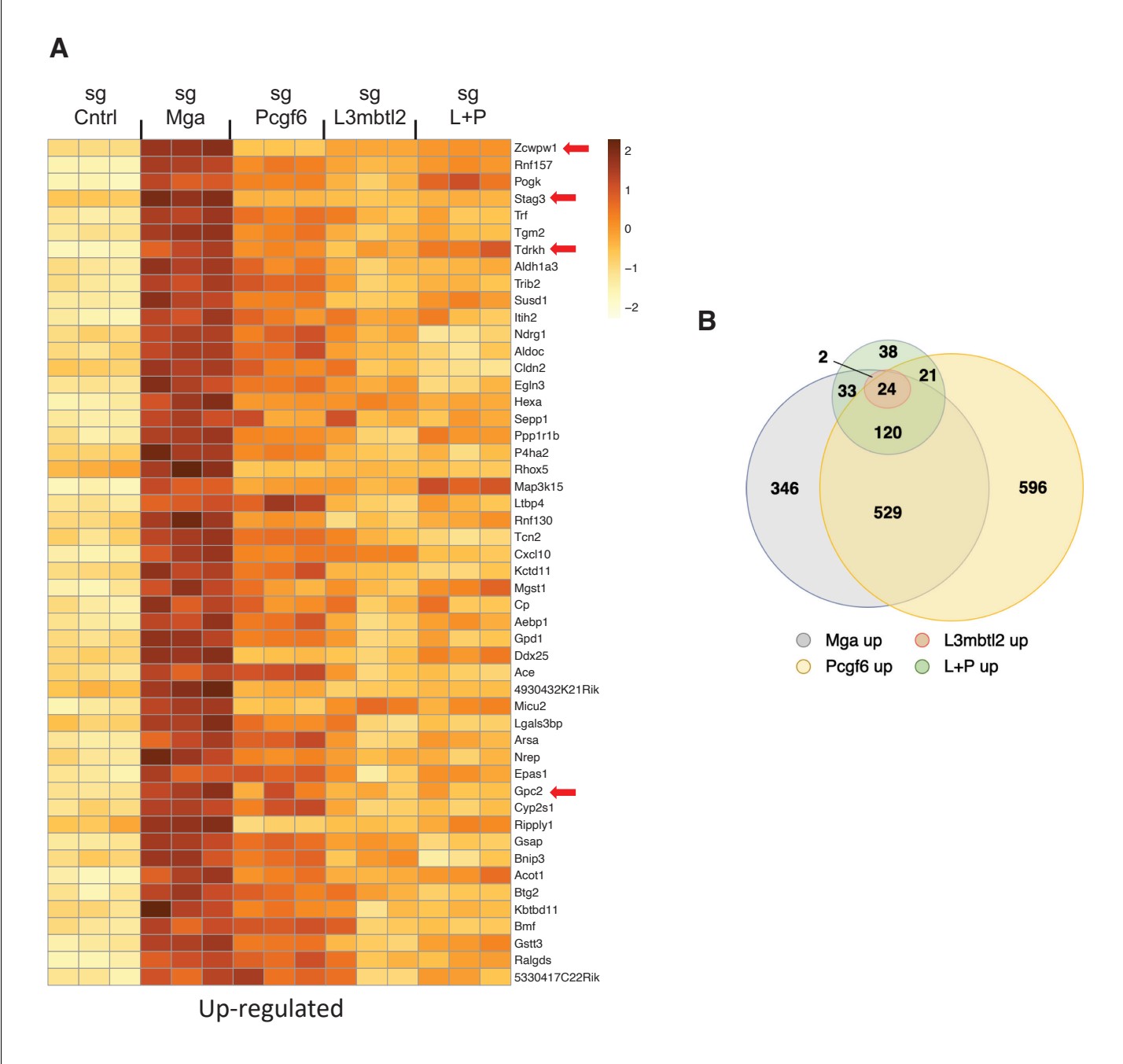

**Figure 6.** PRC1.6 complex subunits contribute to MGA mediated repression. (**A**) Heat map of 50 top upregulated RNAs in KP tumor cell line treated with sgControl, sgMga, sgL3MBTL2, and sgPCGF6 +sgL3MBTL2 as determined by RNA-Seq. Differential expression of RNAs ranked according the adjusted p-value in the sgMga sample, n = 3 for each sgRNA. (**B**) Venn diagram indicate extent of overlap among upregulated genes from the different CRSPR deletions shown in (**A**). Genes with adjusted p-value<0.05 were included.

The online version of this article includes the following figure supplement(s) for figure 6:

**Figure supplement 1.** PRC1.6 complex subunits contribute to MGA mediated activation.

adenocarcinoma lines (*Figure 7—figure supplement 1E,G*). Interestingly, MGA suppression conferred no growth advantage to these lines, in contrast to the phenotypes we observed in models where MGA was inactivated at the tumor initiation stage (*Figure 7—figure supplement 1F,H* compare with *Figure 1B,F, and I*). However, we observed an increase in invasive properties upon MGA

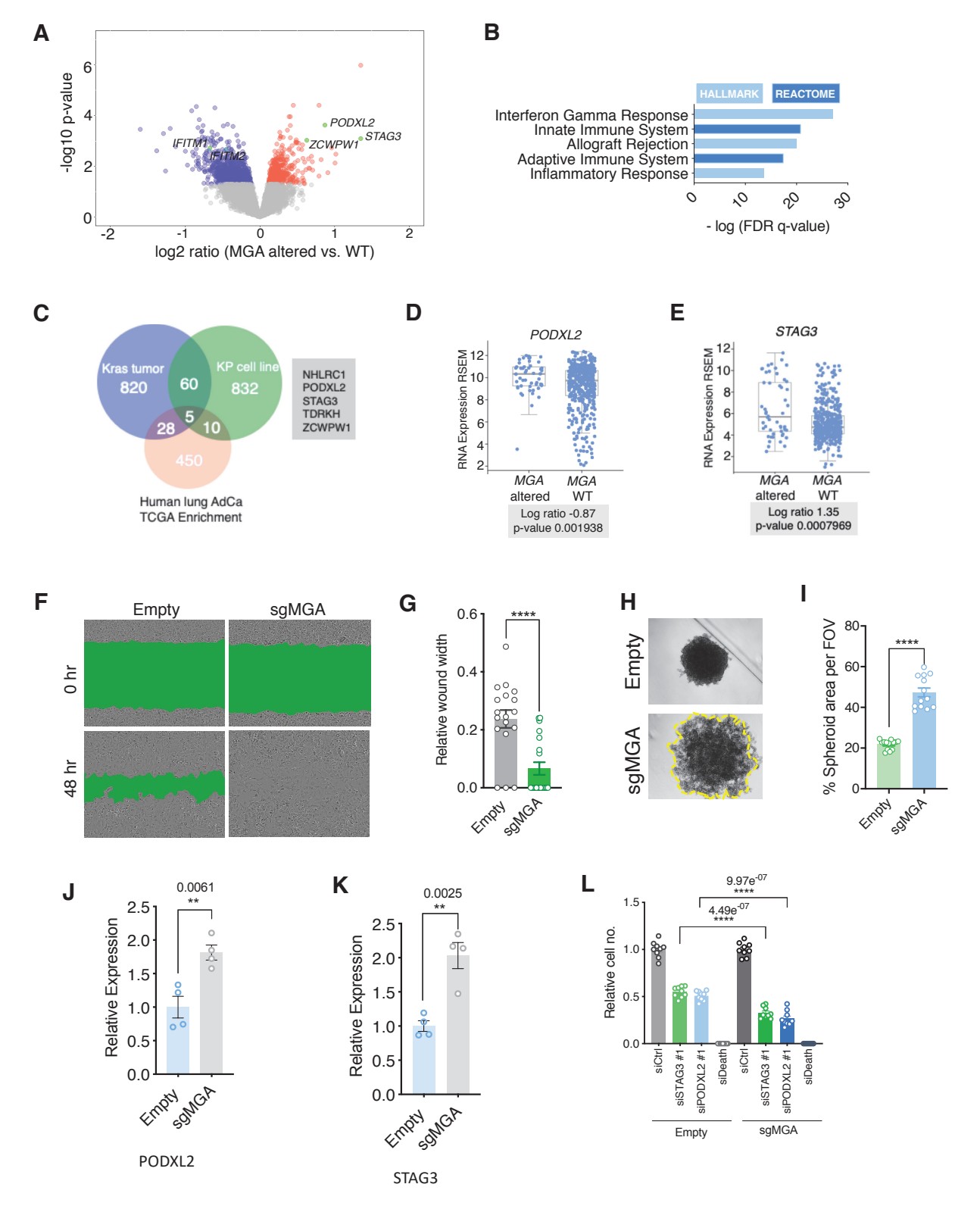

**Figure 7.** MGA loss correlates with activation of *STAG3* and *PODXL2* in human lung cancer and results in a pro-invasive phenotype in vitro. (**A**) Volcano plot of transcripts that are differentially expressed in MGA altered (n = 52) vs. WT (n = 455) from pancancer TCGA lung adenocarcinoma data. Representative genes highlighted in green. (**B**) Hallmark and Reactome analysis of genes downregulated in MGA altered patients vs. non-altered (p-value cutoff < 0.05). (**C**) Venn diagram depicting overlap between genes upregulated in the mouse Kras tumor, KP cell line, and Human lung TCGA

*Figure 7 continued on next page*

*Figure 7 continued*

data with MGA alterations. (D, E) Log ratios of (D) PODXL2 and (E) STAG3 expression in MGA altered vs. WT lung adenocarcinoma patients (TCGA, 2014). (F) Representative wound widths and (G) quantification of wound width at 48 hr in Empty and sgMGA wells (n = 18 Empty, n = 20 sgMGA. ****p=4.77e$^{-05}$). (H) Spheroid formation and (I) quantification in Empty and sgMGA A549 lines (quantification at Day 6, n = 13 spheroids for each from three independent experiments. ****p=4.89e$^{-11}$). (J, K) qPCR for (J) PODXL2 and (K) STAG3 expression in Empty and sgMGA A549 cells (n = 4 replicates for each condition). (L) Cell growth upon siRNA knockdown of STAG3 and PODXL2 in A549 Empty and sgMGA cells (n = 3 replicates from three independent experiments for each group). All p-values calculated using a two-sided Student's t-test unless otherwise noted. Error bars represent SEM.

The online version of this article includes the following source data and figure supplement(s) for figure 7:

Source data 1. MGA loss correlates with activation of *STAG3* AND *PODXL2* in human lung cancer and results in a pro-invasive phenotype in vitro.
Figure supplement 1. Manipulation of MGA in human lung cancer cell lines results in a pro-invasive phenotype and activation of PRC1.6 targets.

inactivation as evidenced by faster migration of MGA-inactivated cells in wound healing assays (*Figure 7F,G*, *Figure 7—figure supplement 1I,J*) and formation of less compact spheroids by A549 sgMGA cells (*Figure 7H,I*). In addition, we observed an up-regulation of PODXL2 and *STAG3* (*Figure 7J,K*). In order to study the importance of these genes in lung cancer cells, we performed siRNA knockdowns of *STAG3* and *PODXL2* in control and sgMGA cells (*Figure 7—figure supplement 1K and L*). Knockdown of either gene led to a substantial reduction in cell growth and this effect was more pronounced in sgMGA cells (*Figure 7L*). Taken together, these results indicate that the changes we observed in mouse models, including the upregulation of MYC and ncPRC1.6 targets upon MGA loss, are consistent with those observed in patient data and human lung cancer lines.

Given that the 301aa region encompassing the DUF domain is critical in recruiting ncPRC1.6 components, we assessed whether knockdown of individual PRC1.6 complex members could phenocopy MGA loss in a human cell line. To this end, we used CRISPR to inactivate L3MBTL2 and PCGF6 in A549 cells (*Figure 7—figure supplement 1M*). Neither loss of L3MBTL2 nor PCGF6 phenocopied the effects seen on migration (*Figure 7—figure supplement 1N,O*) in MGA CRISPR cells. These data suggest that MGA possesses functions independent of ncPRC1.6 , a conclusion consistent with our data on gene expression in KP cells deleted for the same PRC1.6 subunits (*Figure 6*).

## MGA has a tumor suppressive function in colorectal cancer

As mentioned previously, genetic alterations in MGA are prevalent across a broad spectrum of tumors (*Cancer Genome Atlas Network et al., 2018*). To extend our studies, we aimed to elucidate MGA's role in other tumor types with a high frequency of MGA alterations. We were particularly interested in colorectal cancer since GENIE and TCGA data pointed to a significant percentage of MGA alterations in colorectal adenocarcinoma patients (*Figure 8A*). To study MGA's role in colorectal cancer initiation, we utilized a CRISPR-based approach to inactivate MGA in normal human colon organoids and confirmed that MGA levels were decreased in sgMGA organoids compared to those infected with the vector control (*Figure 8B*, *Figure 8—figure supplement 1A*). We confirmed indel formation by sequencing two clonal sgMGA organoids (*Figure 8—figure supplement 1B*). MGA loss was accompanied by a decrease in L3MBTL2 levels, as we had observed in our lung cancer models (*Figure 8B*, *Figure 3B,D*). Inactivation of MGA in colon organoids did not impact growth in 2D culture (*Figure 8—figure supplement 1C*) but clearly accelerated 3D growth over a period of 14 days (*Figure 8C,D*, *Figure 8—figure supplement 1D*). To further characterize changes mediated by MGA that facilitate a pro-growth phenotype, we performed global gene expression profiling comparing control vector and sgMGA transduced organoids. PCA revealed that sgMGA organoid gene expression profiles form a distinct cluster when compared to controls (*Figure 8E*). Importantly, we observed that four of the five genes (*STAG3*, *PODXL2*, *NHLRC1*, and *ZCWPW1*) identified from our analysis in lung cancer models were significantly upregulated upon loss of MGA (*Figure 8F*, *Supplementary file 1e*). Enrichment analysis revealed that there are both common and distinct MGA driven transcriptional changes in colon organoids when compared to our murine lung cancer studies. Similar to the mouse lung adenocarcinoma KP model, the expression of several EMT-related genes was elevated upon MGA loss. Strikingly, we also noted a broad upregulation of E2F regulated, cell cycle and DNA replication associated genes (*Figure 8G*). As in the Kras tumors, inflammation and interferon signaling genes were significantly enriched amongst the downregulated genes

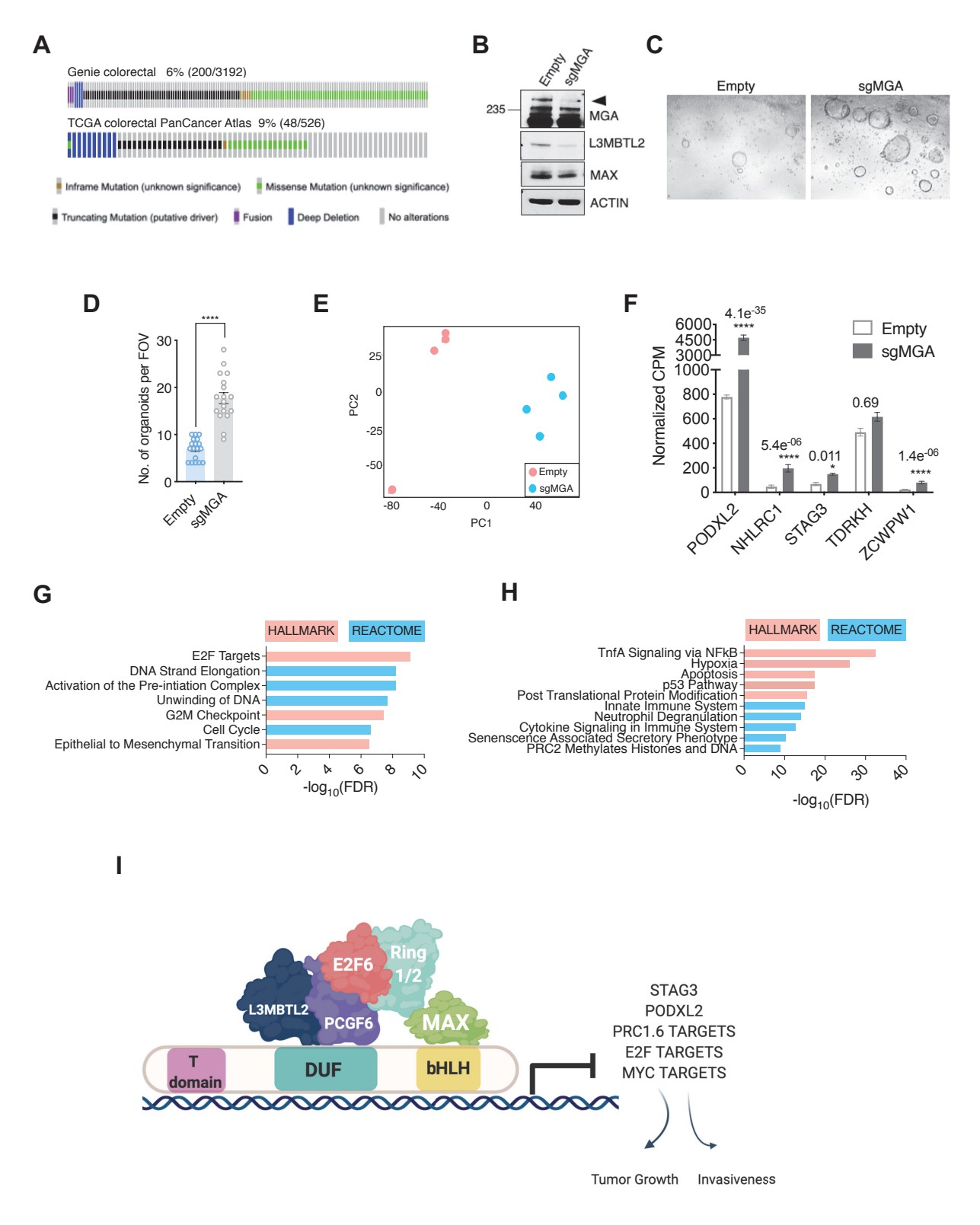

**Figure 8.** MGA has tumor suppressive functions in colorectal cancer. (A) GENIE and TCGA consortium data depicting alterations at the MGA locus in colorectal cancer. (B) Western blot for MGA, L3MBTL2, and MAX in Empty and sgMGA normal colon organoids. (C) Representative images and (D) quantification of normal human colon organoid growth from single cells following MGA inactivation using CRISPR (*p=8.84e$^{-10}$. Two-sided Student's t-test, n = 18 FOV across six biological replicates). (E) PCA plots for Empty and sgMGA organoids (n = 4 biological replicates for each). (F) Normalized

*Figure 8 continued on next page*

*Figure 8 continued*

CPM values for MGA target genes in Empty and sgMGA organoids (n = 4 biological replicates for each). GSEA Hallmark and Reactome analysis for genes (G) upregulated and (H) downregulated upon MGA loss in colon organoids (FDR < 0.05, LFC > 1.5). (I) Proposed mechanism of MGA mediated tumor suppressive effects: MGA acts as a scaffold and stabilizes atypical PRC1.6 members, including L3MBTL2. Under normal conditions, this results in the repression or transcriptional attenuation of thousands of genes. During malignant progression, perturbation of MGA expression leads to the upregulation of growth-promoting and pro-invasive PRC1.6, MYC, and E2F targets in a tissue-specific manner.

The online version of this article includes the following source data and figure supplement(s) for figure 8:

**Source data 1.** MGA has tumor suppressive functions in colorectal cancer.

**Figure supplement 1.** MGA has tumor suppressive functions in colon cancer.

(*Figure 8H*). Pathways unique to colon organoids also scored significantly. For example, several colon stemness associated genes including *LGR5* were upregulated upon MGA loss (*Figure 8—figure supplement 1E*; *Muñoz et al., 2012*; *van der Flier et al., 2009*). Conversely, senescence appeared to be a major process enriched amongst the downregulated genes (*Figure 8H*, *Figure 8—figure supplement 1E*).

Overall, our data in colorectal cancers further extend the concept that MGA functions as a tumor suppressor in multiple tissue types and that normal organoids can serve as an in vitro system for study of early events that are triggered by MGA loss of function.

## Discussion

### Tumor-suppressive roles within the extended MYC network

A recent TCGA pan-cancer analysis found *MYC* paralogs to be amplified in 28% of all samples. Moreover, in the same study, other members of the MYC network such as the presumptive MYC antagonists MXD, MNT, and MGA proteins were found to be inactivated in a subset of tumors (*Cancer Genome Atlas Network et al., 2018*). Arguably, an equilibrium between activating and repressive heterodimers within the network is likely to be critical for tissue homeostasis and can be disrupted in oncogenesis (*Augert et al., 2020*; *Carroll et al., 2018*; *Link et al., 2012*; *Nguyen et al., 2020*). To investigate the functions of putative MYC antagonists more deeply we focused on MGA, the largest bHLHZ protein known to heterodimerize with MAX. Intriguingly, MGA sustains a high rate of inactivating alterations across a wide range of malignancies (*Cancer Genome Atlas Network et al., 2018*). We chose to further investigate MGA function in lung cancer since a substantial percentage of lung adenocarcinoma patients (6–10%) harbor MGA alterations (*Figure 1—figure supplement 1*). Recent studies indicate that MGA function is consistent with it acting as a MYC antagonist and tumor suppressor (*Llabata et al., 2020*; *Reddy et al., 2017*). For example, introduction of MGA into lung adenocarcinoma cells significantly retards their growth (*Llabata et al., 2020*). Here we demonstrate that MGA inactivation dramatically accelerates tumor progression in a murine model of lung adenocarcinoma and triggers oncogenic conversion in human-derived colon tissues. Our work furnishes in vivo evidence for MGA acting as a potent tumor suppressor, identifies genes directly regulated by MGA that are candidates for contributing to oncogenesis and provides model systems for further studies on MGA function.

### MGA functions within the ncPRC1.6 complex

Cancer genome sequencing efforts reveal that in addition to mutations in growth-promoting signaling pathways such as in EGFR and KRAS, chromatin-modifying enzymes and epigenetic regulators such as KMT2D, ARID1A, and TET2 are also frequent targets of genetic alterations (*Kandoth et al., 2013*). Dysregulation of Polycomb repression mediated by PRC2 has been characterized in several cancers (reviewed in *Laugesen et al., 2016*). For example, elevated levels of EZH2, a methyltransferase associated with PRC2, drives tumor progression in mouse models of lung adenocarcinoma through the establishment of a unique super-enhancer landscape (*Zhang et al., 2016*). In addition, studies have established a context-dependent function for canonical PRC1 complex members in malignant transformation (*Koppens and van Lohuizen, 2016*). However, the role of non-canonical PRC1 complexes in malignant cells is poorly understood. Recent reports indicate that the four distinct ncPRC1 complexes (PRC1.1; PRC2/4; PRC3/5; and PRC1.6) function in normal development

through interaction with genomic DNA and recruitment of histone modifying enzymes, including histone methyltransferases, and ubiquitin ligases (*Gao et al., 2012*; *Trojer et al., 2011*). ncPRC1.6 is the only ncPRC1 complex reported to include MAX and MGA. In terms of gene expression, ncPRC1.6 binding has been associated with both permissive transcription and repression (*Ogawa et al., 2002*; *Scelfo et al., 2019*; *Stielow et al., 2018*). Importantly, the ncPRC1.6 complex, through its MGA-MAX mediated DNA binding activity, has been demonstrated to critically control the repression of meiotic genes during both male and female murine germ cell development and to suppress differentiation of embryonic stem cells (*Endoh et al., 2017*; *Hishida et al., 2011*; *Suzuki et al., 2016*). In line with these observations, our expression profiling studies in MGA-deficient tumors and cell lines revealed a significant upregulation of known germ cell related transcripts such as *Stag3*, *Tdrkh*, and *Zcwpw1*. Furthermore, these genes, together with *Podxl2* and *Nhlrc*, comprise a group of five genes whose expression is significantly induced in murine sgMga-Kras, and sgMga-Kras/p53 tumors and in TCGA human lung cancers with MGA alterations (*Figure 7A,C*). Four of these genes were significantly upregulated upon MGA deletion in human colon organoids (*Figure 8F*). In both the lung and colon, we also detected a less marked de-repression of several hundred genes impinging on a diverse array of processes ranging from lipid metabolism to EMT.

Our genomic occupancy analysis in both mouse and human lung cancer lines is entirely consistent with the widespread recruitment of ncPRC1.6 subunits with MGA-MAX binding sites on genomic DNA. We detect overlap between members of the ncPRC1.6 complex (E2F6, L3MBTL2, MAX, and MGA) at several thousand promoters (*Figure 5A*). Furthermore, a high percentage of the genes whose expression is altered upon MGA deletion are directly bound by the MGA-MAX complex. The latter include most of the loci encoding germ cell related proteins (*Figure 5D*). The data suggest that the majority of MGA-bound genes in these cells are ncPRC1.6 regulated. A subset of these genes is also bound by E2F6, consistent with previous studies suggesting that MGA represses at E2F6-Dp1 sites through ncPRC1.6 (*Llabata et al., 2020*; *Ogawa et al., 2002*). We show that loss of MGA results in greatly diminished L3MBTL2 and E2F6 genomic binding (*Figure 5B,C*). Taken together with our observation that MGA stabilizes ncPRC1.6 complex members in tumor cells, our findings strongly implicate MGA as an anchor or scaffold for PRC1.6 even in malignant settings.

The link between ncPRC1.6 activity and the effects of MGA loss on specific gene expression was further investigated by comparing gene expression profiles of Crispr-mediated deletion of *Mga* with deletions of ncPRC1.6 subunits *Pcgf6* and *L3mbtl2* in KP tumor cells. Deletion of either *Pcgf6*, *L3mbtl2,* or bothresults in expression profiles that are similar but not identical to the profiles generated by *Mga* deletion (*Figure 6*; *Figure 6—figure supplement 1*). While the overall direction of change due to *Mga* vs. *Pcgf6/L3mbtl2* deletions is similar, the degree of change is attenuated for many genes. Thus, while *Pcgf6* and *L3mbtl2* contribute to the transcriptional effect consequent to *Mga* deletion, they do not fully account for the degree of change in gene expression. It is likely that other ncPRC1.6 subunits (e.g. E2F6) or factors (e.g. HDACs, CBX3) associated with other regions of MGA (*Figure 3*) play key roles in the transcriptional and physiological response to MGA loss of function.

## The DUF region is critical for MGA's tumor suppressive function

Our mass spectrometric analysis using fragments of MGA revealed that aa1000–1300 are critical for MGA interactions with L3MBTL2. This region encompasses a domain of unknown function known as DUF4801, and several lines of evidence suggest that it is important for mediating tumor suppression. First, addback of a mutant form of MGA lacking this region fails to retard the growth of MGA-deficient human and mouse lung cancer lines (*Figure 4*). Second, multiple truncating mutations in MGA reported in the pan-cancer TCGA database either remove the DUF region entirely or occur within the DUF region (*Figure 4—figure supplement 1*). Lastly, MGA truncations in leukemia due to intronic polyadenylation also occur in this region (*Lee et al., 2018*). The truncated form of MGA produced in this manner acts as a dominant negative and activates MYC and E2F targets. A very recent study reported that a MGA splice variant lacking the bHLHZ region, yet still associating with ncPR1.6, acts as a dominant interfering form during germ cell development (*Kitamura et al., 2020*). Further experiments will be required to determine whether the truncated MGA DUF mutant, as well as other truncated or point mutated forms, promote oncogenesis.

## MGA target gene activity in the context of MYC network and E2F targets

Our ChIP-Seq data show extensive overlap among genes occupied by MYC, MAX, RNA polymerase II, and the MGA-MAX-ncPRC1.6 complex (*Figure 5A,M*), and it is tempting to assume that MGA plays a role as a putative MYC antagonist similar to the MXD/MNT proteins. However, mechanistically, MGA repression, mediated by ncPRC1.6, appears to be distinct from the repressive activity of the MXD/MNT proteins, mediated by Sin3-HDAC. MGA is also unusual in possessing a functional T-box as well as a bHLHZ DNA binding domain (*Hurlin et al., 1999*). A very recent report demonstrates that both of these domains are involved in MGA-induced repression of meiotic genes in murine ES cells (*Uranishi et al., 2020*). It is conceivable that MGA responds to a distinct set of signals and is involved in the regulation of a unique subset of genes relative to other network members. As mentioned above, MGA-PRC1.6 has been demonstrated to play a key role in repression of meiotic gene expression and more generally in the suppression of differentiation and the maintenance of pluripotency (*Endoh et al., 2017*; *Hishida et al., 2011*; *Washkowitz et al., 2015*). These functions are consistent with our observation that meiotic PRC1.6 targets such as *STAG3* and *TDRD1* as well as pro-invasive genes such as *PODXL2* and mesenchymal lineage genes, like *SNAI1*, are induced upon MGA loss in tumors. Overall, this suggests that during development, in non-transformed cells, MGA-PRC1.6 normally represses inappropriate lineage-specific transcripts. Indeed, we observe a robust upregulation of EMT-associated genes, meiotic genes, and colon stem cell markers upon loss of MGA in normal colon organoids (*Figure 8F*, *Figure 8—figure supplement 1*). That at least some of these genes are relevant to human cancer is supported by a recent report that high STAG3 expression in colorectal cancer is associated with metastasis, drug resistance, and poor clinical outcomes (*Sasaki et al., 2021*).

Analysis of our gene expression profiling studies and chromatin occupancy in mouse and human lung cancer models reveals a strong enrichment for E2F4 and E2F6 targets amongst genes bound by and regulated by MGA. In addition, our studies in colon organoids reveal a robust activation of E2F cell cycle gene targets. This is consistent with studies showing that MGA in association with PRC1.6 occupies E2F and MYC responsive genes in resting (G0) cells (*Ogawa et al., 2002*). A large body of evidence implicates a critical role for MYC and E2F cooperation in maintaining normal tissue homeostasis (*Liu et al., 2015*; *Mathsyaraja et al., 2019*; *Pickering et al., 2009*).

Our genomic occupancy studies also reveal that nearly all MGA-bound loci are associated with RNA polymerase II and that the genes upregulated upon MGA deletion display increased polymerase binding (*Figure 5H,J*). This suggests that the presence of MGA may normally limit the extent of polymerase binding at a subset of genes. MYC-bound genes are also associated with RNA polymerase II but in general do not show increased polymerase binding upon MGA loss (*Figure 5L*). Nonetheless, we observed that MYC occupancy increases at promoters of certain genes upregulated upon MGA deletion, including several ncPRC1.6 meiotic gene targets (*Figure 5E,F*, *Figure 5—figure supplement 1B,C*). Analysis of TCGA data suggests that MGA loss in lung adenocarcinoma patients is not frequently accompanied by amplification or highly increased expression of MYC family proteins, although we show that MYC is required for growth and survival of both MGA wild-type and mutant cells (*Figure 5—figure supplement 1D,E*). The role of MYC in MGA-inactivated tumors may involve a limited redistribution of MYC to a subset of promoters or enhancers and needs to be explored in more detail.

Based on our results, we propose that MGA inactivation in pre-malignant settings destabilizes PRC1.6 and consequentially results in unoccupied E-boxes and E2F/Dp1 sites. A combination of invasion of E-boxes by MYC-MAX and/or binding of E2Fs at target loci may account for the net transcriptional activation of these genes and tumor initiation or progression (*Figure 8*). However, our studies do not rule out the possibility of other transcription factors contributing to this phenotype. For example, the T-box factor Brachyury is known to promote EMT and invasiveness in several different tumor types, including lung and hepatocellular carcinoma (*Du et al., 2014*; *Fernando et al., 2010*; *Shah et al., 2017* reviewed in *Chen et al., 2020*). Perhaps MGA mediates repression of EMT related transcripts, in addition to meiotic genes, via its T-box domain (*Uranishi et al., 2020*). Further structure–function studies will enable the delineation of the role of MGA's T-box domain in suppressing tumorigenesis.

Lastly, we note that, in both the lung adenocarcinoma cells and colon organoids, MGA deletion is also accompanied by downregulation of multiple genes. While some of this repression may be indirect, it is nonetheless apparent that direct MGA binding is associated with a subset of the genes downregulated upon MGA loss (*Figure 5D*), which overall are associated with decreased RNA polymerase II binding (*Figure 5K*). In this regard, it is noteworthy that recent reports on ncPRC functions ascribe both activation and repression activities related to ncPRC1.6 subcomplex recruitment with distinct binding and transcriptional activities (*Fursova et al., 2019*; *Scelfo et al., 2019*; *Stielow et al., 2018*). While the genes downregulated upon MGA loss in our lung adenocarcinoma cells did not delineate evident functional pathways, the downregulated genes in the colon organoids were involved in inflammatory and interferon gamma responses (*Figure 8H*, *Figure 8—figure supplement 1E*). The possibility of MGA acting as a transcriptional activator remains to be tested, and further studies will be required to delineate the specific components of the immune microenvironment that are impacted by MGA loss in human malignancies.

In summary, we demonstrate that MGA acts as a bona fide tumor suppressor in vivo and have uncovered a critical role for MGA and the atypical PRC1.6 complex in lung adenocarcinoma. We also provide preliminary evidence for a tumor suppressive role for MGA in colorectal cancer. Although our data is suggestive of a broad tumor suppressive role of MGA and PRC1.6 via antagonizing transcription of MYC and E2F targets, further studies will be required to elucidate the functional relevance of individual MGA-PRC1.6 targets such as STAG3 in tumorigenesis. Given the frequent inactivation of MGA across diverse cancer types, it will also be critical to further characterize MGA- and PRC1.6-mediated transcriptional attenuation in other malignancies with MGA alterations, such as colorectal cancer and diffuse large B-cell lymphoma. This is especially important as MGA, via atypical PRC1.6, likely regulates lineage-specific genes to mediate its tumor suppressor role in a context-dependent manner.

# Materials and methods

**Key resources table**

| Reagent type (species) or resource | Designation | Source or reference | Identifiers | Additional information |
|---|---|---|---|---|
| Strain, strain background (*M. musculus*, male, female) | Kras | The Jackson Laboratory | RRID:IMSR_JAX:008179 | *KrasLSL-G12D/WT* |
| Strain, strain background (*M. musculus*, male, female) | KP (Kras p53) | The Jackson Laboratory | RRID:IMSR_JAX:008462 | *KrasLSL-G12D/WT Trp53Flox/Flox* K and P interbred at Fred Hutch |
| Cell line (human) | A549 | ATCC | CCL-185 RRID:CVCL_0023 | Lung epithelial |
| Cell line (human) | NCI-H23 | ATCC | CRL-5800 RRID:CVCL_1547 | Lung adeno-carcinoma, epithelial |
| Cell line (human) | NCI-H2291 | ATCC | CRL-5939 RRID:CVCL_1546 | Lung adeno-carcinoma, epithelial |
| Cell line (human) | LOU-NH-91 | Kind gift from Montse Sanchez- Cespedes | RRID:CVCL_2104 | Lung squamous cell carcinoma |
| Cell line (human) | 91T | Kind gift From McGarry Houghton | Reference: Stabile et al. 2002 | Non-small cell lung cancer |
| Cell line (human) | NCI-H1975 | ATCC | CRL-5908 RRID:CVCL_1511 | Lung adeno-carcinoma, epithelial |
| Cell line (human) | DLD-1 | ATCC | CCL-221 RRID:CVCL_0248 | Colorectal adeno-carcinoma, epithelial |
| Recombinant DNA reagent (human) | pcDNA3-FLAG- His-MGA | Yuzuru Shiio | | Plasmid to express full length MGA isoform2 Available from the Eisenman Lab |

*Continued on next page*

*Continued*

| Reagent type (species) or resource | Designation | Source or reference | Identifiers | Additional information |
|---|---|---|---|---|
| Recombinant DNA reagent (human) | pCDH-FLAG-His- MGA | This paper | | Lentiviral construct to express full-length MGA Available from the Eisenman Lab |
| Recombinant DNA reagent (human) | pCDH-FLAG-His- MGAΔDUF | This paper | | Lentiviral construct to express MGA lacking the aa1003- 1304 Available from the Eisenman Lab |
| Recombinant DNA reagent (*M. musculus*) | LentiCRISPRv2Cre | Addgene (Gift from Feldser lab) | | |
| Recombinant DNA reagent (*M. musculus*) | LentiCRISPRv2Cre sgMga1 | This paper | Insert (sgMga1 gRNA sequence): TAAGTGGAATGGTCGTTGGT | Lentiviral construct to transfect and express sgRNA Available from the Eisenman Lab |
| Recombinant DNA reagent (*M. musculus*) | LentiCRISPRv2Cre sgMga3 | This paper | Insert (sgMga3 gRNA sequence): TCAGAATTTTCAATATACGC | Lentiviral vector to express sgRNA Available from Eisenman Lab |
| Recombinant DNA reagent (human) | lentiCRISPRv2 PURO | Addgene (Gift from Feng Zhang) | | Lentiviral vector to express sgRNA |
| Recombinant DNA reagent (human) | lentiCRISPRv2 sgMGA | This paper | Insert (sgMGA gRNA sequence): CTATGCATCGTTACCTGC CG | Lentiviral vector to express sgRNA Available from the Eisenman Lab |
| Recombinant DNA reagent (human) | lentiCRISPRv2 sgPCGF6 | This paper | Insert (sgPCGF6 gRNA sequence): GGTATGAAGACATTCTGTGA | Lentiviral vector to express sgRNA Available from the Eisenman Lab |
| Recombinant DNA reagent (human) | lentiCRISPRv2 sgL3MBTL2.1 | This paper | Insert (sgL3MBTL2.1 gRNA sequence): CCGGAGTTATAACAGCAGTG | Lentiviral vector sgRNA Available from the Eisenman Lab |
| Recombinant DNA reagent (*Mus musculus*) | LentiCRISPRv2sgL3mbtl2 | This Paper | Insert (sgL3mbtl2 gRNA sequence): CCGGAGTTACAACAGCAGTG | Lentiviral construct to transfect and express sgRNA Available from the Eisenman Lab |
| Recombinant DNA reagent (*Mus musculus*) | LentiCRISPRv2sgPcgf6 | This Paper | Insert (sgPcgf6 gRNA sequence): GGTATGAAGACACTCTGTAA | Lentiviral construct to transfect and express sgRNA Available from the Eisenman Lab |
| Recombinant DNA reagent (*Mus musculus*) | LentiCRISPRv2sgMyc | This Paper | Insert (sgMyc gRNA sequence): TGTCCCCGAGCCGCCGCTC | Lentiviral construct to transfect and express sgRNA Available from the Eisenman Lab |
| Recombinant DNA reagent (human) | lentiCRISPRv2 sgL3MBTL2.1 | This paper | Insert (sgL3MBTL2.2 gRNA sequence): TTCACTTGACTCCTCCAGAT | Lentiviral vector to express sgRNA |
| Recombinant DNA reagent (human) | pLKO non-silence | SIGMA Mission shRNA | | Lentiviral vector to express shRNA |
| Recombinant DNA reagent (human) | pLKO shMGA | SIGMA Mission shRNA | TRCN0000237941 (ID RNAi consortium) | Lentiviral vector to express shRNA |

*Continued on next page*

*Continued*

| Reagent type (species) or resource | Designation | Source or reference | Identifiers | Additional information |
|---|---|---|---|---|
| Antibody | MGA (rabbit, polyclonal) | Novus Biological | Catalog no. NBP1-94031 | WB (1:500 in 3%BSA), IF (acetone fixation and 1:100) |
| Antibody | MGA (mouse, monoclonal) | Developed at Fred Hutch | | WB (1:500 in 5% milk) |
| Antibody | MGA (rabbit, polyclonal) | Kind gift from Suske Lab (Univ. Marburg) | | ChIP, CUT, and RUN (1:100) |
| Antibody | Ki67 (rabbit, polyclonal) | Abcam | Catalog no. ab16667 | IF and IHC (1:200) |
| Antibody | MECA32 (mouse, monoclonal) | DSHB | Catalog no. MECA-32-S | IHC (1:100) |
| Antibody | MYC (rabbit, polyclonal) | Cell Signaling | Catalog no. 13987S | ChIP, CUT and RUN (1:200) |
| Antibody | MAX (rabbit, polyclonal) | Proteintech | Catalog no. 10426–1-AP | WB (1:1000), ChIP, CUT and RUN |
| Antibody | L3MBTL2 (rabbit, polyclonal) | Active Motif | Catalog no. 39569 | WB (1:1000), ChIP, CUT and RUN |
| Antibody | RNA polymerase II (all phosphoisomers) | Active Motif | Catalog no. 39097 | ChIP (1:100) |
| Antibody | E2F6 (rabbit, polyclonal) | Abcam | Catalog no. ab53061 | WB (1:1000), ChIP |
| Antibody | PCGF6 (rabbit, polyclonal) | Proteintech | Catalog no. 24103–1-AP | WB (1:1000) |
| Antibody | Rabbit IgG | Cell Signaling | Catalog no. 2729 | ChIP |
| Antibody | GFP (rabbit, monoclonal) | Cell Signaling | Catalog no. 2956S | CUT and RUN |
| Sequence-based reagent (human) | siSTAG3 | Qiagen | siRNA (Catalog no. SI00734440) | |
| Sequence-based reagent (human) | siPODXL2 | Qiagen | siRNA (Catalog no. SI04142495) | |
| Sequence-based reagent (human) | siMYC | Qiagen | siRNA (Catalog no./ID: 1027416) | FlexiTube GeneSolution GS4609 for MYC(human) |
| Sequence-based reagent (human) | siDeath | Qiagen | siRNA (Catalog no./ID: 1027299) | ALLSTARS HS Cell Death siRNA |
| Sequence-based reagent (human) | siCtrl/siCTRL | Qiagen | siRNA (Catalog no./ID: 1027281) | ALLSTARS Negative Control siRNA |
| Software | Biorender | | Biorender.com | Used for illustrations |

## Mouse models and generation of cell lines

All mice used in the study were housed and treated according to the guidelines provided by the Fred Hutch Institutional Animal Care and Use Committee. The *Kras G12D* LSL and *Trp53* floxed alleles were maintained on a C57BL6 background. Heterozygous *Kras G12D* LSL/+ alone (termed Kras) or in combination with *Trp53* fl/fl homozygous alleles (termed KP) were used for experiments. Mice 2 months or older were utilized for intratracheal instillation of lentiviral particles in a BSL2 facility (*DuPage et al., 2009*). For generation of KP mouse lines, tumors were collected using sterile surgical instruments from endpoint KP mice. Tumors were mechanically disaggregated using a wide bore pipette tip and cultured in DMEM supplemented with 10% FBS and antibiotics. Cells were passaged until tumor cells outgrew stromal contaminants (immune cells, fibroblasts, and endothelial cells) and used for subsequent downstream applications.

## Cloning, transfection, and lentiviral transduction

For in vivo CRISPR, two sgRNA-targeting mouse Mga (sgMga#1 and sgMga#3) were designed and cloned into the lentiCRISPRv2cre vector (*Walter et al., 2017*). See Key resources table for sequences. Virus was generated by the Fred Hutch Cooperative Center for Excellence in Haematology Viral Vector core and titered using qPCR. For in vitro studies in human lung cancer lines, a guide RNA against MGA was cloned into the lentiCRISPRv2 puro vector (kind gift from Feng Zhang). Viral supernatant was generated via transfection of 293FT or 293TN cells with lentiCRISPR v2 constructs and packaging vectors pPAX2 and pVSV-G using lipofectamine 2000 (Invitrogen). Supernatant was cleared of cellular debris using a 0.45 µM filter, and target cells were transduced with viral supernatant containing 4 µg/ml polybrene final concentration. Cell were grown in 10 µg/ml puromycin 3 days post-transduction. For addback of MGA or MGAΔDUF to mouse KP lines, cells were transfected on chamber slides with pCDH-MGA-FLAG or pCDH-MGAΔDUF-FLAG using lipofectamine 2000 (Invitrogen). Cells were harvested 3 days post-transfection and fixed for immunofluorescent staining. The cloning strategy was also used for cloning a sgRNA into the lentiCRISPRv2 construct with guide RNAs against human *PCGF6*, human *L3MBTL2*, mouse *Pcgf6*, mouse L3mbtl2, and mouse *Myc*. See Key resources table for sequences.

## Cell lines and reagents

All cell lines employed in experiments cited in this manuscript were either derived by the authors or obtained from verified sources and were routinely tested for and found to be free of mycoplasma. Human cell lines were deidentified. Mouse KP and KP-sgMga lung tumor cell lines were generated as described above and cultured in DMEM with 10% FBS. Human non-small cell lung cancer lines NCI-H23, NCI-H2291, NCI-H2347, NCI-H1975, and 91T were grown in RPMI-1640 with 10% FBS. Human squamous cell carcinoma line LOU-NH-91 was maintained in RPMI-1640 supplemented with 20% FBS. A549 human lung adenocarcinoma, DLD1 human colorectal adenocarcinoma, 293FT and 293TN cells were maintained in DMEM with 10% FBS.

## Human colon organoid culture and maintenance

Organoid cultures were established as previously described (*Guo et al., 2019*). Briefly, organoids were plated in Matrigel (Corning #354230), and IntestiCult organoid growth media (StemCell #6010) was supplemented with 10 µM of ROCK Inhibitor Y27632 (ATCC ACS-3030) for 1 week post-establishment. For lentiviral transduction, supernatant was added to individual wells of cells in ultra-low attachment plates. The plate was centrifuged at 600 g at 32℃ for 1 hr. Plate was removed from centrifuge and incubated for 37℃ for 4 hr. Cells were washed in cold 1× PBS containing 10 µM of ROCK Inhibitor Y27632 and pelleted. Cells were then plated in a 10 µl ring of Matrigel in a 96-well tissue culture plate. After Matrigel hardened, 100 µl of IntestiCult containing 10 µM of ROCK Inhibitor Y27632 was added to each well. Two days post-plating 4 µg/ml of puromycin (Sigma #P8833) was added to culture media and refreshed every 48 hr. Single-cell organoid colonies were established via single cell dissociation using TrypLE Express enzyme, described above. Single cells were plated in 10 µl Matrigel rings in a 96-well plate and expanded. Organoid viability was measured using RealTime-Glo MT Cell Viability Assay (Promega #G9711), following manufacturer's protocol. Single-cell organoids were plated at a density of $5 \times 10^3$ cells per well in a 96-well, 2D culture format, described previously (*Thorne et al., 2018*). Bright field images were acquired using Nikon Eclipse E800 fluorescent microscope.

## DNA extraction and sequencing for indels

To confirm indels, cell pellets and tumors were incubated in a Proteinase K containing digestion buffer overnight at 55℃. DNA was either isolated using 25:24:1 phenol:chloroform:isoamylalcohol followed by ethanol precipitation or extracted using the GeneJet genomic DNA purification kit (ThermoFisher). For human colon organoids, cells were dissociated into single cells and DNA was isolated using the Purelink Genomic DNA isolation kit (Invitrogen). PCR was performed using high-fidelity Phusion DNA polymerase (ThermoFisher) with primers flanking the region targeted by sgMga#1 and sgMga#3 (for mouse tumors and cell lines) and sgMGA (for human colon organoids). PCR products obtained were run on a 0.8% agarose gel and amplicons to confirm the presence of one amplicon corresponding to the correct size. PCR DNA was purified using a PCR purification kit

(Qiagen) and sequenced using one of the sgMga#1 sgMga#3 (mouse) and sgMGA (human) primers. Sequence information was input for ICE analysis (Synthego) to compute percent indel formation (https://ice.synthego.com), and Clustal Omega (https://www.ebi.ac.uk/Tools/msa/clustalo/) was used to align sequences.

## Immunostaining and western blots

Cells were cultured on 8-well chamber slides for immunocytochemistry. Briefly, cells were fixed in either 4% formaldehyde or 1:1 methanol:acetone, blocked, and incubated with primary antibodies followed by Alexa fluor conjugated secondary antibodies or Alexa 488-conjugated phalloidin (for Actin staining). Chambers were then removed and slides mounted using Prolong Gold anti-fade with DAPI. Immunohistochemistry was performed on 5 μm thick paraffin embedded mouse lung sections. Following heat-induced antigen retrieval, sections were incubated with primary antibodies followed by Alexa conjugated secondary antibodies. Sections were mounted in Prolong Gold anti-fade with DAPI (Invitrogen). Images were acquired using either TissueFaxs (Tissugnostics), Nikon E800, Deltavision Eclipse, Olympus Fluoview confocal, or Zeiss confocal microscopes. Image analysis and intensity measurements were performed using ImageJ.

For protein gels, whole-cell lysates were prepared using RIPA buffer with protease and phosphatase inhibitors. They were then reduced in NuPage LDS buffer (Invitrogen), and a wet transfer was performed prior to western blotting. For histone blots, acid extracts were made following manufacturer (Abcam) recommendations. Refer to Key resource table for list of antibodies used.

## RNA isolation, qPCR, and RNA-sequencing

For all applications, RNA was isolated using Trizol (ThermoFisher Scientific) according to manufacturer's recommendations or processed using the Direct-zol RNA Miniprep kit (Zymoresearch). For qPCR, cDNA was synthesized from 500 ng to 2 μg of total RNA using the Revertaid cDNA synthesis kit (ThermoFisher). A Biorad iCycler was using for SYBR green-based quantitative PCR. Refer to *Supplementary file 2a,2b* for primer sequences. For RNA sequencing experiments, Kras-frozen tumors or KP cell lines were used. Following RNA isolation, total RNA integrity was checked using an Agilent 4200 TapeStation and quantified using a Trinean DropSense96 spectrophotometer (Caliper Life Sciences). Libraries prepared using the either the TruSeq RNA Sample Prep v2 kit (Illumina) with 500 ng input RNA or Ultra RNA Library Prep Kit for Illumina (New England Biolabs). An Illumina HiSeq 2500 was utlilized to performed paired-end sequencing. Reads that did not pass Illumina's base call quality threshold were removed and then aligned to mm10 mouse reference genome using TopHat v2.1.0. Counts were generated for each gene using htseq-count v0.6.1p1 (using the 'intersection-strict' overlapping mode). Genes that did not have at least one count/million in at least three samples were removed. Data was normalized and comparisons conducted using the exact test method in edgeR v3.18.1. Gene set enrichment analysis was performed using Hallmark and Reactome datasets on mSigDB (*Liberzon et al., 2015*; *Subramanian et al., 2005*). Heatmaps were generated using Morpheus (https://software.broadinstitute.org/morpheus).

## Tandem affinity purification of MGA-interacting proteins and mass spectrometry

Twenty 15 cm plates of 293 T cells were transfected with pcDNA3-FLAG-His-MGA, MGA(967–1300), or MGA(2153–2856); 48 hr after transfection, the cells were lysed in TN buffer (10 mM Tris pH 7.4/ 150 mM NaCl/1% NP-40/1 mM AEBSF/10 μg/ml aprotinin/10 μg/ml Leupeptin/1 μg/ml Pepstatin A/ 20 mM sodium fluoride). The lysate was incubated with Ni-NTA agarose (Qiagen) and FLAG-His-MGA and its interacting proteins were collected by centrifugation, washed three times with TN buffer, and eluted with 50 mM sodium phosphate buffer pH 8.0/150 mM NaCl/250 mM imidazole. The eluted sample was immunoprecipitated with anti-FLAG antibody (M2, Sigma-Aldrich), the immunoprecipitate was eluted with FLAG peptide (Sigma-Aldrich), and the eluted protein sample was processed with an Amicon Ultra 0.5 3 k centrifugal filter device (Millipore) for concentration and buffer exchange to 50 mM Tris pH 8.5. Proteins were digested at 37°C overnight with trypsin (Promega; 1:10, enzyme/substrate) in the presence of 10% acetonitrile. The resulting tryptic peptides were analyzed by HPLC-ESI-tandem mass spectrometry on a Thermo Fisher LTQ Orbitrap Velos Pro mass spectrometer. The Xcalibur raw files were converted to mzXML format and were searched against

the UniProtKB/Swiss-Prot human protein database (UniProt release 2016_04) using X! TANDEM CYCLONE TPP (2011.12.01.1 - LabKey, Insilicos, ISB). Methionine oxidation was considered as a variable modification in all searches. Up to one missed tryptic cleavage was allowed. The X! Tandem search results were analyzed by the Trans-Proteomic Pipeline, version 4.3. Peptide/protein identifications were validated by the Peptide/ProteinProphet software tools (*Keller et al., 2002*; *Nesvizhskii et al., 2003*). The mass spectrometry proteomics data have been deposited to the ProteomeXchange Consortium via the PRIDE partner repository (*Perez-Riverol et al., 2019*) with the dataset identifier PXD025930.

## Genomic occupancy analyses

For genomic occupancy studies in mouse KP lines, conventional crosslinked ChIP-seq was performed (*Skene and Henikoff, 2015*) on one *Mga*-inactivated line and one control line. Chromatin IPs were done using antibodies against MGA, MAX, MYC, L3MBTL2, and E2F6. Libraries were generated using the NEB Ultra II kit (E7645S, NEB), followed by $50 \times 50$ paired-end sequencing on an Illumina HiSeq 2500 instrument. For human cell lines, CUT and RUN was utilized to determine genomic occupancy. Cells were bound to Concanavalin A beads (86057–3, Polysciences Inc), followed by permeabilization using a digitonin containing buffer. One million cell aliquots were then incubated overnight with antibodies against MGA, MAX, MYC, and L3MBTL2. An automated CUT and RUN protocol was followed for library preparation (*Janssens et al., 2018*), and libraries were sequenced using $25 \times 25$ paired-end sequencing on an Illumina HiSeq 2500 instrument. Five to 10 million reads were obtained per antibody. Refer to Key resources table for list of antibodies used.

For ChIP-seq experiments, sequences were aligned to the mm10 reference genome assembly using Bowtie2. For CUT and RUN experiments, alignment was performed to hg38. Library normalization was utilized for ChIP-Seq experiments and spike-in normalization was performed using yeast DNA spike-in for CUT and RUN studies. Peak calling was performed using MACS at different thresholds. Peaks called were further processed using bedtools plus a combination of custom R scripts defining genome position and the GenomicRanges R package. For MGA, peaks were identified as being associated with a gene if they were within + or – 5 kb from the TSS. For CUT and RUN experiments, the intersection of MGA peak calls from two independent experiments following IgG subtraction was utilized to obtain a gene list. The R package ggplot2 or ngs.plot (*Shen et al., 2014*) were used to generate heatmaps for genomic binding. Volcano plots were generated using the R package ggplot2. Enrichr was utilized to overlap peak calls with existing ENCODE and ChEA data. De novo and known motif enrichment for sequence specificity was determined using HOMER (*Heinz et al., 2010*).

## Analysis of TCGA data

Data from the TCGA PanCanAtlas project was downloaded from https://gdc.cancer.gov/about-data/publications/pancanatlas (https://www.cancer.gov/tcga). Classification of somatic MGA mutations as homozygous or heterozygous was obtained from the ABSOLUTE-annotated MAF file containing the output from the ABSOLUTE method (*Carter et al., 2012*), which infers cancer cell ploidy and tumor purity from next-generation sequencing of bulk cancer DNA. MGA-mutant lung adenocarcinoma (LUAD) samples were divided into two groups, heterozygous and homozygous, based on categorization in the homozygous output field. Variants with silent or intronic mutations (n = 10) or ambiguous MGA absolute annotations (n = 3) were excluded from this analysis. 39 LUAD tumor samples had a total of 44 MGA alterations (n = 11 [homozygous], n = 33 [heterozygous]).

Gene expression data of MYC-family genes in lung adenocarcinomas from the TCGA PanCanAtlas dataset were downloaded from cBioportal.org. mRNA expression z-scores relative to all samples (log RNA Seq V2 RSEM) were compared between *MGA*-mutant and non-mutant samples using unpaired two-tailed t-tests.

## Cell growth, spheroid, and migration assays

For 2D growth curves, cells were seeded in 96-well flat bottom dishes (Corning) and imaged at fixed time intervals using either an Incucyte S3 or Incucyte zoom (Essen Bioscience). Percent confluence was used as a measure of cell growth. For 3D spheroid based assays, 5000 A549 cells were plated per well in ultra-low attachment plates and centrifuged at low speed (800 rpm) for 9 min. Phase-

contrast images were acquired using an inverted scope. Spheroid area was calculated using ImageJ. For wound healing assays, cells were plated to reach full confluence prior to the start of the assay. Imagelock plates (Essen Bioscience) were utilized. Cells were pre-treated with mitomycin C (2.5–5 µg/ml) to retard cell growth, and scratch wounds were made using a Woundmaker (Essen Bioscience). Wound closure was monitored at fixed time intervals, and end point wound width or confluence was used as metrics for cell migration.

## Acknowledgements

We would like to thank the members of the Eisenman and MacPherson labs for scientific input and sharing reagents. We are grateful to Guntram Suske, Bastian Stielow, Akihiko Okuda, and Peter Hurlin for their generous gifts of key reagents, to Arnaud Augert and Patrick Carroll for helpful discussions and a critical reading of the manuscript. We also acknowledge the Fred Hutch Genomics and Bioinformatics, Scientific Imaging, Experimental Histopathology, and Small Animal Imaging Shared Resources for their excellent technical support. Scientific computing infrastructure was supported by ORIP (S10OD028685). We are also grateful for technical help provided by the Cooperative Center for Excellence in Hematology (CCEH) at Fred Hutch. We thank the University of Texas Health Science Center at San Antonio Institutional Mass Spectrometry Laboratory (Director: Dr. Susan Weintraub) for mass spectrometry analysis.

## Additional information

### Competing interests

Robert N Eisenman: Scientific Advisory Board Member: Kronos Bio Inc.; Shenogen Pharma Beijing. No overlap with the present study. The other authors declare that no competing interests exist.

### Funding

| Funder | Grant reference number | Author |
| --- | --- | --- |
| National Cancer Institute | R35 CA231989 | Robert N Eisenman |
| National Cancer Institute | RO1 CA200547 | David MacPherson |
| National Cancer Institute | U01 CA235652 | David MacPherson |
| Cancer Prevention and Research Institute of Texas | RP160487 | Yuzuru Shiio |
| Cancer Prevention and Research Institute of Texas | RP160841 | Yuzuru Shiio |
| Cancer Prevention and Research Institute of Texas | RP190385 | Yuzuru Shiio |
| William and Ella Owens Medical Research Foundation | Research Grant | Yuzuru Shiio |
| The Brotman Baty Institute | Pilot Award | William Grady |
| Cottrell Family Fund | Research Grant | William Grady |
| Geiger Family Foundation | Research Grant | William Grady |
| Listwin Fund | Research Grant | William Grady |
| Hartwell Foundation | Innovation Fund Pilot Grant | Robert N Eisenman |
| National Institutes of Health | R50 CA233042 | Ming Yu |
| National Institutes of Health | P30 CA0544174 (UTexas MS facility) | Yuzuru Shiio |
| Fred Hutchinson Cancer Research Center | Translational Data Science Fellowship | Sitapriya Moorthi |
| National Cancer Institute | R37 CA252050 | Alice H Berger |
| Prevent Cancer Foundation | Devereaux Outstanding | Alice H Berger |

| | Investigator Award | |
| --- | --- | --- |
| Lung Cancer Research Foundation | Research grant | William Grady |
| Seattle Foundation | Research grant | Alice H Berger |
| Fred Hutchinson Cancer Research Center | Endowed Chair | Alice H Berger |
| Fred Hutchinson Cancer Research Center | ORIP S10OD028685 | Brian Freie |
| National Cancer Institute | P50 CA228944 | Alice H Berger |

The funders had no role in study design, data collection and interpretation, or the decision to submit the work for publication.

### Author contributions

Haritha Mathsyaraja, Conceptualization, Investigation, Writing - original draft, Writing - review and editing; Jonathen Catchpole, Emily Eastwood, Ekaterina Babaeva, Jessica Ayers, Nan Wu, Kumud R Poudel, Investigation; Brian Freie, Michael Geuenich, Data curation, Formal analysis; Pei Feng Cheng, Investigation, Methodology; Ming Yu, A McGarry Houghton, Conceptualization; Sitapriya Moorthi, Formal analysis, Visualization; Amanda Koehne, Formal analysis; William Grady, Conceptualization, Supervision, Project administration; Alice H Berger, Conceptualization, Formal analysis, Supervision; Yuzuru Shiio, Conceptualization, Data curation, Formal analysis, Investigation, Methodology; David MacPherson, Conceptualization, Formal analysis, Funding acquisition, Project administration, Writing - review and editing; Robert N Eisenman, Conceptualization, Supervision, Funding acquisition, Writing - original draft, Project administration, Writing - review and editing

### Author ORCIDs

David MacPherson ⓘ https://orcid.org/0000-0003-3729-907X
Robert N Eisenman ⓘ https://orcid.org/0000-0002-0274-9846

### Ethics

Human subjects: Patient-derived, de-identified, normal organoids were isolated from subjects from the ColoCare Consortium (Approved by Fred Hutchinson Cancer Center IR6407).
Animal experimentation: This study was performed in strict accordance with the recommendations in the Guide for the Care and Use of Laboratory Animals of the National Institutes of Health. All of the animals were handled according to approved institutional animal care and use committee (IACUC) protocol (#50783) of the Fred Hutchinson Cancer Research Center. Every effort was made to minimize pain and suffering.

### Decision letter and Author response

Decision letter https://doi.org/10.7554/eLife.64212.sa1
Author response https://doi.org/10.7554/eLife.64212.sa2

## Additional files

### Supplementary files

• Supplementary file 1. RNA-Seq , ChIP-Seq and Cut & Run data used in this study. (a) Kras tumor RNA-Seq data. (b) KP cell line RNA-Seq data. (c) KP cell line ChIP-Seq called peaks. (d) A549 CUT and RUN peaks called. (e) Human colon organoid RNA-Seq data.

• Supplementary file 2. Real-time mouse and human PCR primer sequences used in this study. (a) Real-time mouse primers used in the study. (b) Real-time human primers used in this study.

• Transparent reporting form

## Data availability

High throughput sequence data has been submitted to GEO. The accession numbers are as follows: mouse Kras tumor RNA Seq (GSE161609), mouse KP cell line RNA Seq (GSE161606), Human colon organoid RNA Seq (GSE161543), mouse KP cell line ChIP Seq (GSE161541), human A549 CUT&RUN (GSE161539).

The following datasets were generated:

| Author(s) | Year | Dataset title | Dataset URL | Database and Identifier |
|---|---|---|---|---|
| Mathsyaraja H, Catchpole J, Eastwood E, Babaeva E, Geuenich M, Cheng PF, Freie B, Ayers J, Yu M, Wu N, Poudel KR, Koehne A, Grady W, Houghton AM, Shiio Y, MacPherson DP, Eisenman RN | 2020 | RNA sequencing comparing mouse lung cancer cell lines derived from Kras G12D Trp53 -/- and Kras G12D Trp53 -/- sgMga tumors | https://www.ncbi.nlm.nih.gov/geo/query/acc.cgi?acc=GSE161606 | NCBI Gene Expression Omnibus, GSE161606 |
| Mathsyaraja H, Catchpole J, Eastwood E, Babaeva E, Geuenich M, Cheng PF, Freie B, Ayers J, Yu M, Wu N, Poudel KR, Koehne A, Grady W, Houghton AM, Shiio Y, MacPherson DP, Eisenman RN | 2020 | RNA sequencing comparing mouse Kras G12D lung tumors that have WT or inactivated Mga via CRISPR | https://www.ncbi.nlm.nih.gov/geo/query/acc.cgi?acc=GSE161609 | NCBI Gene Expression Omnibus, GSE161609 |
| Mathsyaraja H, Catchpole J, Eastwood E, Babaeva E, Geuenich M, Cheng PF, Freie B, Ayers J, Yu M, Wu N, Poudel KR, Koehne A, Grady W, Houghton AM, Shiio Y, MacPherson DP, Eisenman RN | 2020 | CUT&RUN studies comparing occupancy of MAX, MGA, MYC and L3MBTL2 occupancy in Empty vector and sgMGA transduced A549 lung adenocarcinoma cells | https://www.ncbi.nlm.nih.gov/geo/query/acc.cgi?acc=GSE161539 | NCBI Gene Expression Omnibus, GSE161539 |
| Mathsyaraja H, Catchpole J, Eastwood E, Babaeva E, Geuenich M, Cheng PF, Freie B, Ayers J, Yu M, Wu N, Poudel KR, Koehne A, Grady W, Houghton AM, Shiio Y, MacPherson DP, Eisenman RN | 2020 | Genomic occupancy of MYC and members of the PRC1.6 complex (MGA, E2F6, MAX, L3MBTL2) in MGA WT (Empty) and MGA depleted (sgMga) KrasG12D Trp53 -/- mouse lung cancer lines | https://www.ncbi.nlm.nih.gov/geo/query/acc.cgi?acc=GSE161541 | NCBI Gene Expression Omnibus, GSE161541 |
| Mathsyaraja H, Catchpole J, Eastwood E, Babaeva E, Geuenich M, Cheng PF, Freie B, Ayers J, Yu M, Wu N, Poudel KR, | 2020 | RNA sequencing comparing human colon organoids transduced with Empty control vs. sgMGA (guide RNA targeting MGA) | https://www.ncbi.nlm.nih.gov/geo/query/acc.cgi?acc=GSE161543 | NCBI Gene Expression Omnibus, GSE161543 |

| | | | | |
|---|---|---|---|---|
| Koehne A, Grady W, Houghton AM, Shiio Y, MacPherson DP, Eisenman RN | | | | |
| Mathsyaraja H, Catchpole J, Eastwood E, Babaeva E, Geuenich M, Cheng PF, Freie B, Ayers J, Yu M, Wu N, Poudel KR, Koehne A, Grady W, Houghton AM, Shiio Y, MacPherson DP, Eisenman RN | 2021 | MGA interactome in 293T cells using affinity purification and mass spectrometry | https://www.ebi.ac.uk/pride/archive/projects/PXD025930 | PRIDE, PXD025930 |
| Mathsyaraja H, Catchpole J, Eastwood E, Babaeva E, Geuenich M, Cheng PF, Freie B, Ayers J, Yu M, Wu N, Poudel KR, Koehne A, Grady W, Houghton AM, Shiio Y, MacPherson DP, Eisenman RN | 2021 | RNA sequencing comparing in-vitro lentiviral CRISPR-Cas9 knockdown of Non-Canonical Polycomb Repressive Complex 1.6 members from a Kras G12D Trp53 -/- background mouse lung cancer cell line, ChIP-Seq files for E2F6, L3MBTLK2, MGA, IgG, RNA polymerase II in MGA WT and sgMGA KP cells | https://www.ncbi.nlm.nih.gov/geo/query/acc.cgi?acc=GSE175838 | NCBI Gene Expression Omnibus, GSE175838 |

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
