## [Decision Letter]

Thank you for submitting your article "Loss of MGA mediated Polycomb repression promotes tumor progression and invasiveness" for consideration by *eLife*. Your article has been reviewed by 3 peer reviewers, including Martin Eilers as the Reviewing Editor and Reviewer #1, and the evaluation has been overseen by Kevin Struhl as the Senior Editor.

The reviewers have discussed the reviews with one another and the Reviewing Editor has drafted this decision to help you prepare a revised submission.

The reviewers agreed that the paper provides strong in vivo data for a tumor-suppressive role for Mga in lung carcinogenesis. The authors are convincing that MGA is important in oncogenesis. We note here that MGA is highly understudied (~200 publications) in and of itself despite its involvement with the MYC network for oncogenesis (~41,000 publications at the current time). Given a protein of 3000 amino acids, the number of potential protein partners and PTMs that might modify its tumor suppressor functions are staggering. However, the reviewers also noted that a previous paper has addressed the same topic and the novelty of the data presented here needs to better explained and additional experiments are needed to strengthen and expand the new aspects.

There are several loosely articulated stories here, but the reader does not come away with a clear take-home message. No particular oncogenic pathway is intensively explored or elucidated. MGA seems to be quite a general tumor suppressor, so is there a reason to focus experiments on lung vs. colon adenocarcinomas, or are these convenient models to explore general MGA action? The PRC1.6 story is interesting, but it is not integrated into a general description of MGA action. We think the authors buried the lede here; the take home message should be that there are multiple, perhaps independent mechanisms for tumor suppression embedded in MGA; these mechanisms should be clearly enumerated and the evidence for each supported. As presented, there is little attempt to separate or coordinate the various anti-oncogenic functions and contributions of the different domains of MGA-T-box, bHLH, or DUF480. Just considering the bHLH domain, there are multiple possibilities for MGA to disturb the MYC-network and suppress oncogenesis. The actin protrusions seem to be related to DUF480 but unrelated to PRC1.6; they probably relate with the observations about migration, wound healing and EMT (and so may also be relevant for T-box functions). I would suggest a serious re-write to identify, organize and reinforce their story(ies).

1. It seems that the investigation of publicly available datasets is essentially identical to the Schaub et al.. analysis and not new data. If the authors want to maintain this, they would need to better explain what is new. One important piece of information that seems to be missing is whether the mutations are homozygous or heterozygous. So data on MGA and MYC protein expression in human tumors would greatly strengthen this part.

Inspections of the MGA mutations on cBio portal reveals broad tissue specificity of tumors, well beyond lung and colon adenocarcinomas-in view of this range there is really no need to justify the choice of tumors, only to be wary whether differences in expression of MGA targets truly reflect MGA versus the tissue as the source of a non-pathogenic difference. Looking at the distribution of the mutations within MGA, I disagree with the assertion that DUF480 is a hotspot. The only lukewarm to hot spot in the protein seems to be the bHLH region. The other mutations are rather evenly distributed throughout the protein. This does not mean that they are not pathogenic, more likely it indicates a broad range of contributory motifs and domains. I suspect that there is a lot of interesting and important biology throughout these 3000 residues.

2. Conceptually, one would like to know whether tumor development in an MGA-delete situation depends on MYC. One would also like to know whether the polycomb complex that is assembled by MGA is tumor-suppressive. Therefore, the authors should perform a similar analysis as they did for MGA (introduce sgRNAs into the lung models) and score the phenotypes they get. Both experiments could be done in cell lines established from this model and either in vitro (that would allow a mechanistic analysis, e.g. RNA seq) or upon re-transplantation. This would also prevent simply reporting negative results.

3. The interpretation of the VENN diagram and the heatmaps in Figure 5A,B is somewhat uncertain. If one plots these for MYC, occupancy often simply parallels occupancy by RNAPII, so essentially being bound by MYC simply says the promoter is open/active. Is this the case for MGA and its complex partners? Or is there a specificity in binding? The authors should do RNAPII ChipSeqs in these cells, preferentially +/- MGA, and then show these alongside (and plot a correlation between MYC, RNAPII and MGA occupancy).

Along these lines, it is hard to understand how one obtain the extreme p-values shown in figure 5E and 5H, I would challenge this. If the authors want to maintain this, they should not use ENCODe data, but simply determine what genes are active in the cells (e.g. what promoters are bound by RNAPII) and then use those as background list and calculate P-values for overlap between MYC, MAX and E2F6.

Based on the description, the ChIPSeq analyses are not spike-normalized and I could not find information about the number of repeats. If it is n=1, the authors need to find a way to exclude that the differences are due to experimental variation.

4. The authors use the term "Empty" vector for their sgMGA -CRISPR-cas-Cre lentivirus without the sgRNA. This is confusing-what they mean is minus-sg; the vector is hardly "empty". While it would have been nice to have a non-targeting sgRNA just to ensure that no part of the phenotype reflects off-target or non-specific effects of CRISPR-CAS or CRE-expression, at least they should make clear in the text what is the difference between the vectors.

5. The methods describing the observation and quantitation of the actin protrusions are a bit sparse. The interpretation relating DUF480 with actin protrusions and with gene expression separate fromPRC1.6 is hard to follow and sparsely supported.

6. Contrary to what is stated in the manuscript, to my eye MAX is reduced in LOU-NH-91.

7. The KP mouse models are driven by activated RAS called RASV12G but not MYC. Is MYC amplified in these models? I was struck by the finding in Figure 5 and comment that whether MGA is present or not, MYC will drive tumor growth. Some lung adenocarcinomas acquire MYC amplification. Do the human lines used in the experiments have MYC amplified?

It is unclear to me whether MAG loss and MYC amplification are mutually exclusive or correlated? The authors should comment as to whether MGA is required or not in MYC amplified tumors.

8. Does MGA-MAX compete with MYC-MAX to bind the E boxes, and if so, would enforced expression of MYC in the tumors reverse the phenotype? It would be good for the authors to define MAG as the MAX-Gene associated protein.

9. While the title states that "Loss of MGA mediated polycomb repression..", I would add atypical or non-canonical polycomb to the title to distinguish is from the other better known PCR2 and PCR1 complexes.

10. The authors use PRC1.6 rather than PCFG6-PRC1 used by Llabata. The authors should make sure in the text that is refers to the same complex.

11. Figure 4 panels 4B and 4, MGA antibody shows multiple bands. A cleaner blot will help. Is MGA expressed as spliced variants or is the antibody dirty?

---

## [Author Response]

The reviewers agreed that the paper provides strong in vivo data for a tumor-suppressive role for Mga in lung carcinogenesis. The authors are convincing that MGA is important in oncogenesis. We note here that MGA is highly understudied (~200 publications) in and of itself despite its involvement with the MYC network for oncogenesis (~41,000 publications at the current time). Given a protein of 3000 amino acids, the number of potential protein partners and PTMs that might modify its tumor suppressor functions are staggering. However, the reviewers also noted that a previous paper has addressed the same topic and the novelty of the data presented here needs to better explained and additional experiments are needed to strengthen and expand the new aspects.There are several loosely articulated stories here, but the reader does not come away with a clear take-home message. No particular oncogenic pathway is intensively explored or elucidated. MGA seems to be quite a general tumor suppressor, so is there a reason to focus experiments on lung vs. colon adenocarcinomas, or are these convenient models to explore general MGA action? The PRC1.6 story is interesting, but it is not integrated into a general description of MGA action. We think the authors buried the lede here; the take home message should be that there are multiple, perhaps independent mechanisms for tumor suppression embedded in MGA; these mechanisms should be clearly enumerated and the evidence for each supported. As presented, there is little attempt to separate or coordinate the various anti-oncogenic functions and contributions of the different domains of MGA-T-box, bHLH, or DUF480. Just considering the bHLH domain, there are multiple possibilities for MGA to disturb the MYC-network and suppress oncogenesis. The actin protrusions seem to be related to DUF480 but unrelated to PRC1.6; they probably relate with the observations about migration, wound healing and EMT (and so may also be relevant for T-box functions). I would suggest a serious re-write to identify, organize and reinforce their story(ies).

The reviewers point out that our paper provides strong and compelling in vivo data for a tumor-suppressive role for Mga in carcinogenesis. While they note that MGA is relatively understudied, they point out that the recent paper by Llabata et al. also addressed aspects of MGA’s tumor suppressive and DNA binding activities and the reviewers ask us to explain the novelty of our paper.

As noted, a major focus of our paper is that it provides and validates a mouse model in

which we delete MGA and demonstrate its tumor suppressive activity. The

experiments in Llabata et al., including the biological assays and the ChIP-Seq,

were done by *overexpressing* MGA in cells which already express endogenous

MGA. Therefore, all of their data monitor the consequences of overexpression of

MGA, a situation without clear biological relevance. In the experiments reported in

our paper, we delete MGA. Therefore, our molecular data refer to a comparison

between MGA null and the same cells expressing endogenous MGA. This is

important since MGA is a tumor suppressor and its loss of function is what is crucial

biologically, as we show here or the first time in our lung adenocarcinoma model.

In addition, by deleting MGA we were able to show that its loss corresponds to an

increase in a core set of target genes previously associated with PRC1.6.

Furthermore, we show that members of this core group are directly bound by the MGA-MAX PRC1.6 complex and are relevant to the proliferation of tumors that lack MGA (Figure 5, Figure 7L).

The PRC1.6 complex has been known to be associated with MGA since at least

2012 as indicated in our references cited. Llabata et al. confirmed that result in somatic cells as do we, but we extend this data to show that PRC1.6 subunits are associated with MGA through the DUF4801domain of MGA. This is the first identification of the interface between PRC1.6 and MGA. It is important and relevant because multiple frame shift mutants in MGA have the consequence of deleting this region in a wide range of tumor types including lung cancers (Figure 4 -Sup 1B). In our revised paper we cite Llabata as well as earlier reports relating to MGA multiple times and describe in greater detail the differences with our data (line 91+).

Regarding our choice of experimental systems (lung and colon) the reviewer is correct in suggesting that we were influenced by a combination of having available a well-established mouse model for lung adenocarcinoma as well as the ongoing organoid cultures from human colon. Both systems offer the possibility of assessing early events in oncogenic conversion. Furthermore, TCGA data indicates that LUAD is second (following UVEC) in terms of prevalence of MGA mutations. In the Introduction (line 58+) we cite the previous reports that led us to choose these models.

The reviewers point out the potentially “staggering” complexity of MGA and the possibility that it has myriad activities potentially linked to functions within its three described regions (T box, DUF 4801, and bHLHZ) and suggest a re-write of our text to clarify what we know about the structure-function relationships related to MGA. In the revised paper we have reorganized and rewritten many sections to increase clarity and have underscored both in the results and discussion what functions can and cannot be attributed to the various regions of MGA. We can only agree that there is “… a lot of interesting and important biology…” associated with MGA and see our paper as providing a jumping off point for further exploration of this “highly understudied” tumor suppressor.

1. It seems that the investigation of publicly available datasets is essentially identical to the Schaub et al.. analysis and not new data. If the authors want to maintain this, they would need to better explain what is new. One important piece of information that seems to be missing is whether the mutations are homozygous or heterozygous. So data on MGA and MYC protein expression in human tumors would greatly strengthen this part.Inspections of the MGA mutations on cBio portal reveals broad tissue specificity of tumors, well beyond lung and colon adenocarcinomas-in view of this range there is really no need to justify the choice of tumors, only to be wary whether differences in expression of MGA targets truly reflect MGA versus the tissue as the source of a non-pathogenic difference. Looking at the distribution of the mutations within MGA, I disagree with the assertion that DUF480 is a hotspot. The only lukewarm to hot spot in the protein seems to be the bHLH region. The other mutations are rather evenly distributed throughout the protein. This does not mean that they are not pathogenic, more likely it indicates a broad range of contributory motifs and domains. I suspect that there is a lot of interesting and important biology throughout these 3000 residues.

MGA was barely mentioned in the Schaub et al. TCGA paper and was lumped together with MNT and the MXDs. In the present manuscript our point was to highlight the pervasiveness of MGA loss and focus on two tumor types where MGA loss of function is apparent and also include and compare TCGA and Genie-derived data.

In response to the reviewer’s request for more information we have collaborated with Alice Berger and Sita Moorthi (now listed as co-authors) genomic analysis experts who have further analyzed the TCGA data and particularly focused on MGA mutations in lung adenocarcinomas (LUAD) that would be expected to lead to inactivation of the protein through loss of functional domains (e.g. frame shifts and nonsense mutations, all of which lead to truncation of MGA. We have omitted missense mutations since their effects cannot be readily predicted). The available data indicates that all of the truncating mutations in LUAD would be predicted to delete either the T-box, the DUF4801 region, or the bHLHZ or all three (see revised Figure 4- Figure supplement 1B). It was based on this information that we decided to investigate the DUF region in the proteomics experiment (line 253+). Moreover, the available data indicates that 25% of the functional mutations are homozygous. We note that heterozygous mutations may be consequential as there is recent evidence from Okuda’s lab suggesting that, in germ cell development, an alternatively spliced form of MGA can act as a dominant interfering form (Kitamura et al). Please see our comments on (line 580) in the revised paper.

Further analysis of the TCGA data revealed that only one of the LUAD samples with truncated MGA displayed amplification of MYCL and none showed MYCN or MYC amplification, while 3 tumors with MGA missense mutations showed MYC amplifications. Turning to RNA expression levels, the TCGA data show no significant difference in MYC family gene expression in MGA mutants compared to LUAD without MGA alteration (data included in Figure 5-figure supp.1D). Lastly, LUADs with MGA mutations were mutually exclusive with gain or loss of MAX copy number or expression changes, as expected since MAX loss would be expected to inactivate MGA E-box binding. While some LUAD tumor lines have high MYC, in general MYC amplification is not strongly associated with LUAD, in contrast with SCLC where all 3 Myc family members are frequently amplified. This is described in the revised paper starting at line 324.

2. Conceptually, one would to know whether tumor development in an MGA-delete situation depends on MYC. One would also like to know whether the polycomb complex that is assembled by MGA is tumor-suppressive. Therefore,the authors should perform a similar analysis as they did for MGA (introduce sgRNAs into the lung models) and score the phenotypes they get. Both experiments could be done in cell lines established from this model and either in vitro (that would allow a mechanistic analysis, e.g. RNA seq) or upon re-transplantation. This would also prevent simply reporting negative results.

To address the reviewer's request we have now utilized CRISPR to inactivate Myc, Mga, L3mbtl2, Pcgf6, and L3mbtl2+Pcgf6 in tumor derived KP cell lines and verified relevant loss of expression changes in these genes (Figure 6 supp1). To assess potential effects on cellular behavior we carried out proliferation as well as invasion assays. MYC deletion results in growth arrest or apoptosis in the KP and the KP-sgMGA cells (Figure 5- Figure supp 1F) consistent with the MYC and MAX inhibitor data already included in the paper (Figure S5G,H). Deletion of MGA, or PRC1.6 complex subunits had no significant effect on proliferation rate relative to KP controls. In the spheroid assay the sgMGA cells showed marginally greater outgrowth than controls but the sgPRC1.6 had reduced outgrowth compared to controls. These very subtle effects on cell behavior are not entirely surprising given that the KP cells are already derived from advanced lung tumors at a late stage of tumor progression. The optimal experiment would be to generate the individual deletions in the initial murine model to address their role in tumor initiation – but given the time and depth of analysis required, we cannot realistically include it in this paper.

Turning to the RNA-seq data we compared differentially expressed genes in KP cells upon individual deletion of Mga, L3mbtl2, Pcgf6, and L3mbtl2+Pcgf6 cells. In figure 6 we show heat maps and Venn diagrams demonstrating highly significant overlap among genes upregulated upon MGA loss and those upregulated by individual deletion of L3mbtl2 and Pcgf6. Most notably we find that genes associated with meiotic recombination and repair are among the most differentially upregulated upon Mga deletion as well as by deletion of Pcgf6 or L3mbtl2 (e.g. Stag3, Tdrkh, Zcwpw1) as well as genes involved in EMT. These findings support the notion of a critical role for the PRC1.6 complex in gene regulation by MGA. At the same time it is clear that deletion of key PRC1.6 factors do not fully recapitulate the effect of Mga loss. This underscores the notion that other regions of MGA contribute to the deletion phenotype. This data has been included in revised Figure 6. We have now added a separate section devoted to what we understand about the role of MYC in MGA deleted tumors (starts at line 335 and in the Discussion starting at line 584+).

3. The interpretation of the VENN diagram and the heatmaps in Figure 5A,B is somewhat uncertain. If one plots these for MYC, occupancy often simply parallels occupancy by RNAPII, so essentially being bound by MYC simply says the promoter is open/active. Is this the case for MGA and its complex partners? Or is there a specificity in binding? The authors should do RNAPII ChipSeqs in these cells, preferentially +/- MGA, and then show these alongside (and plot a correlation between MYC, RNAPII and MGA occupancy).Along these lines, it is hard to understand how one obtain the extreme p-values shown in figure 5E and 5H, I would challenge this. If the authors want to maintain this, they should not use ENCODe data, but simply determine what genes are active in the cells (e.g. what promoters are bound by RNAPII) and then use those as background list and calculate P-values for overlap between MYC, MAX and E2F6.Based on the description, the ChIPSeq analyses are not spike-normalized and I could not find information about the number of repeats. If it is n=1, the authors need to find a way to exclude that the differences are due to experimental variation.

Assessing RNAPII, Myc and Mga occupancy is an excellent suggestion and is something that we had planned to do as a close follow-up to the present study. Nonetheless we have now carried out these experiments and included the data in our revised paper (Figure 5). The data shown are from ChIP-seq and the Venn diagram numbers are derived from the 2 experiments. We have included a new Venn diagrams showing the overlap between MYC, MGA and RNAPII as well as TSS +/-2kb line plots and box plots indicating the concurrence of RNAPII and MGA binding in the +/- MGA conditions. As evident from the data, there is a high degree of overlap between occupancy by MGA, MYC, and RNAPII. We also note a significant increase in RNAPII occupancy to genes upregulated upon MGA deletion (Figure 5J). Downregulated genes and MYC bound genes display a modest decrease in binding (Figure 5K). As requested, we deleted the Encode-based categories from Figure 5. These findings are also discussed in the results and in the Discussion section (line 584).

4. The authors use the term "Empty" vector for their sgMGA -CRISPR-cas-Cre lentivirus without the sgRNA. This is confusing-what they mean is minus-sg; the vector is hardly "empty". While it would have been nice to have a non-targeting sgRNA just to ensure that no part of the phenotype reflects off-target or non-specific effects of CRISPR-CAS or CRE-expression, at least they should make clear in the text what is the difference between the vectors.

We employed the term ‘empty’ to highlight the fact that it was indeed minus sgRNA, something that is not infrequent in the existing literature. Nonetheless we take the reviewer's point that it is ambiguous and have now ensured that it is clearly defined it in the text, as suggested.

5. The methods describing the observation and quantitation of the actin protrusions are a bit sparse. The interpretation relating DUF480 with actin protrusions and with gene expression separate fromPRC1.6 is hard to follow and sparsely supported.

We have removed the data on actin protrusions.

6. Contrary to what is stated in the manuscript, to my eye MAX is reduced in LOU-NH-91.

We pointed out the lack of a strong effect on MAX levels because previous work from us and others has indicated that MAX loss is associated with increased MYC protein degradation. Therefore, the presence of MAX would be consistent with an effect on MYC gene expression rather than protein turnover. In response to the reviewer’s comment, we have measured the levels of MAX by densitometry as shown in Author response image 1 (normalized to actin loading). In the revised text we conclude that feedback within the network may affect MYC and MAX levels.

7. The KP mouse models are driven by activated RAS called RASV12G but not MYC. Is MYC amplified in these models? I was struck by the finding in Figure 5 and comment that whether MGA is present or not, MYC will drive tumor growth. Some lung adenocarcinomas acquire MYC amplification. Do the human lines used in the experiments have MYC amplified?It is unclear to me whether MAG loss and MYC amplification are mutually exclusive or correlated? The authors should comment as to whether MGA is required or not in MYC amplified tumors.

We have not directly measured MYC levels in untransformed and KRAS transformed lung epithelial cells. However, activation of the RAS pathway has been known for many years to result in increased MYC protein stability and this is likely to be the case in the KRAS tumors. Importantly, MYC mRNA, protein, and overall genomic occupancy are not measurably different in the KRAS/p53 tumor derived cells whether or not MGA is deleted. This is consistent with the TCGA data which do not show consistent amplification or elevated expression of MYC in LUAD that correlates with MGA inactivating mutations (see Figure supplement 5D). In fact, amplification of MYC is not very common in LUAD (in sharp contrast to SCLC). Nonetheless there are some LUAD lines with amplified MYC (the human H23 line does have amplification of MYC while the A549 lacks MYC amplification). Our data indicate that the endogenous levels of MYC in both KP and KP-sgMga lines are required for proliferation and survival. It is conceivable that high MYC levels may trigger apoptosis in the absence of MGA. Having said all this it is important to note that the expression of MYC is critical for the growth of most cells including the KP and the KP-sgMGA cells (see Figure 5 supplement 1 E-G). Our future experiments will address the role of MYC and other transcription factors as a consequence of MGA loss.

8. Does MGA-MAX compete with MYC-MAX to bind the E boxes, and if so, would enforced expression of MYC in the tumors reverse the phenotype? It would be good for the authors to define MAG as the MAX-Gene associated protein.

In our paper reporting the identification and initial characterization of MGA (Hurlin et al. 1999 doi: 10.1093/emboj/18.24.7019) we noted that equivalent amounts of the GST-linked bHLHZ regions of MYC and MGA bound roughly equal amounts of MAX and E-box containing DNA oligonucleotides. Rigorous binding affinities have not been determined and would certainly be worthwhile carrying out for MGA as well as other network factors. While not definitive in terms of binding strength, our ChIP analyses indicate that MGA and MYC and other network factors can occupy the same regions of DNA.

In that same 1999 paper we identified MGA as one of a group of novel genes and proteins that heterodimerize with MAX. We named it MGA to: (i) distinguish it from the other group of MAX dimerizing proteins (MAD or MXD proteins) and (ii) to underscore that it is by far the largest of the MAX network proteins. We note in the present paper that it should be pronounced "mega" to reflect this fact (line 59).

9. While the title states that "Loss of MGA mediated polycomb repression..", I would add atypical or non-canonical polycomb to the title to distinguish is from the other better known PCR2 and PCR1 complexes.

Title: at the reviewer's suggestion we have changed the title. In addition we have now referred to this complex as ncPRC1.6 to make it clear throughout the paper that this is a non-canonical complex.

10. The authors use PRC1.6 rather than PCFG6-PRC1 used by Llabata. The authors should make sure in the text that is refers to the same complex.

In our opinion, Llabata's use of PCGF6-PRC1 is unusual and somewhat confusing. PRC1.6 or ncPRC1.6 has been consistently used to refer to this complex by multiple publications over the last decade and we prefer to use this nomenclature.

11. Figure 4 panels 4B and 4, MGA antibody shows multiple bands. A cleaner blot will help. Is MGA expressed as spliced variants or is the antibody dirty?

There are MGA splice variants and there are also likely to be non-specific bands in this blot.